# On the Robustness and Generalization Ability of Building Footprint Extraction on the Example of SegNet and Mask R-CNN

**Muntaha Sakeena [1]**, **Eric Stumpe [1]**, **Miroslav Despotovic [2]**, **David Koch [2]** and **Matthias Zeppelzauer [1,*]**

[1] Institute of Creative Media Technologies, St. Pölten University of Applied Sciences, Campus-Platz 1, A-3100 St. Pölten, Austria
[2] Institute for Energy, Facility & Real Estate Management, University of Applied Sciences Kufstein Tirol, Andreas Hofer-Strasse 7, A-6330 Kufstein, Austria
* Correspondence: matthias.zeppelzauer@fhstp.ac.at

**Abstract:** Building footprint (BFP) extraction focuses on the precise pixel-wise segmentation of buildings from aerial photographs such as satellite images. BFP extraction is an essential task in remote sensing and represents the foundation for many higher-level analysis tasks, such as disaster management, monitoring of city development, etc. Building footprint extraction is challenging because buildings can have different sizes, shapes, and appearances both in the same region and in different regions of the world. In addition, effects, such as occlusions, shadows, and bad lighting, have to also be considered and compensated. A rich body of work for BFP extraction has been presented in the literature, and promising research results have been reported on benchmarking datasets. Despite the comprehensive work performed, it is still unclear how robust and generalizable state-of-the-art methods are to different regions, cities, settlement structures, and densities. The purpose of this study is to close this gap by investigating questions on the practical applicability of BFP extraction. In particular, we evaluate the robustness and generalizability of state-of-the-art methods as well as their transfer learning capabilities. Therefore, we investigate in detail two of the most popular deep learning architectures for BFP extraction (i.e., SegNet, an encoder–decoder-based architecture and Mask R-CNN, an object detection architecture) and evaluate them with respect to different aspects on a proprietary high-resolution satellite image dataset as well as on publicly available datasets. Results show that both networks generalize well to new data, new cities, and across cities from different continents. They both benefit from increased training data, especially when this data is from the same distribution (data source) or of comparable resolution. Transfer learning from a data source with different recording parameters is not always beneficial.

**Keywords:** building footprint extraction; satellite image segmentation; building detection; comparative study; robustness evaluation

## 1. Introduction

The extraction of building footprints refers to precise pixel-accurate detection and tracing of building contours in an orthogonal view of the earth surface [1]. Building footprint (BFP) extraction therefore plays a key role in many applications, such as spatial simulation, 3D building modeling, urban planning, disaster management, damage assessment, and population estimation [2–4]. Respective methods in the scientific literature focus on semiautomatic and automatic extraction approaches using both open and commercial data, such as satellite images, LIDAR data, or digital surface models (DSM). Notable progress in the research in this respect has been achieved in the last decade through the use of supervised machine learning methods employing planimetric building polygons or masks as ground truth into the learning process. While developed approaches showed high performance on the detection of buildings, the design of a method for robust and precise detection of

building boundaries is still an open research topic [5]. To enable comparability of the methods, numerous benchmark datasets have been introduced, such as INRIA [6], AIRS [7], and ISPRS [8] datasets. A limitation of these datasets is that they typically focus on a specific region (e.g., one or several distinct cities) and are thus only representative of a limited area. Thus, models trained from one of these datasets are prone to a bias towards the dataset. Given the high heterogeneity of urban and suburban regions as well as building patterns and styles across different countries and continents (construction types, building density, land cover, etc.), results on such benchmark datasets tell little about the robustness and generalization ability of individual methods.

In the presented study, we investigate the robustness of popular state-of-the-art neural segmentation architectures, namely SegNet [9] and Mask R-CNN [10], which are frequently used for segmentation tasks and represent the core of many BFP extraction approaches. We have selected SegNet and Mask R-CNN for two reasons. First, these networks represent popular backbones for many more advanced recent building footprint extraction approaches. Second, they represent two highly complementary approaches towards segmentation, i.e., SegNet is a one-stage segmentation net with an encoder–decoder architecture, and Mask R-CNN is a two-stage segmentation network which first detects object bounding boxes and then segments them.

In our study, we employ a large-scale high-resolution (30 cm) dataset incorporating images from 24 different locations across Europe and North America (see Table 1). The dataset thereby covers multiple structurally different urban and suburban regions with different building density and type of development (residential, industrial etc.). For better comparability, we further evaluate the approaches using the INRIA and ISPRS datasets on which most BFP extraction methods to date have been evaluated. Our study focuses on the following research questions:

- RQ1: How does training set size affect BFP extraction performance?
- RQ2: How well do the BFP extraction networks generalize to previously unseen cities?
- RQ3: Does transfer learning help in BFP extraction?
- RQ4: How well do the BFP extraction networks generalize to cities from different continents?
- RQ5: Is BFP extraction robust across different settlement/building structures?
- RQ6: How well do the BFP extraction networks generalize to differently dense areas?

Ultimately, the study allows us to conduct a direct comparison of the two different network architectures (encoder–decoder vs. object detection architecture) of SegNet and Mask R-CNN in terms of robustness, transfer learning capabilities and generalization ability for BFP extraction, which is important to support design decisions in practical applications. By answering the above questions, our study tries to derive guidelines for practical application in terms of network architecture, pretraining, fine-tuning, and transfer learning and highlight potential limitations of the employed model architectures.

The results of our study show that both networks in principle generalize well to new data (from previously seen as well as from completely new cities) and across cities from different continents. Furthermore, they improve their segmentations when additional training data become available, making them suitable for online learning settings. This benefit is high when the additional data are from the same distribution (data source) or of comparable resolution. In transfer learning experiments from a data source with deviating recording settings, this benefit was not always observed; on the contrary, performance also dropped in some cases. Our study also shows that the generalization ability across different settlement structures, building densities, and sizes is rather low, which emphasizes the importance of assuring heterogeneity and diversity when creating the training set.

**Table 1.** Overview of the compiled GBDX dataset: European (EU) and North American (NAM) cities in the dataset together with the number of tiles and patches generated from the tiles for BFP extraction.

| EU Cities | Tiles | Patches | NAM Cities | Tiles | Patches |
|---|---|---|---|---|---|
| Athens | 13 | 832 | Raleigh | 24 | 1536 |
| Berlin | 7 | 448 | Santa | 36 | 2304 |
| Madrid | 7 | 448 | Houston1 | 10 | 640 |
| Oslo | 14 | 896 | Peoria | 25 | 1600 |
| Vienna | 15 | 960 | Philadelphia | 23 | 1472 |
| Copenhagen | 53 | 3392 | Detroit1 | 20 | 1280 |
| Hamburg | 22 | 1408 | Houston2 | 19 | 1216 |
| London | 44 | 2816 | San Jose | 23 | 1472 |
| Munich | 45 | 2880 | Detroit | 20 | 1280 |
| Zagreb | 17 | 1088 | Vancouver | 9 | 576 |
| Amsterdam | 15 | 960 | | | |
| Brussels | 10 | 640 | | | |
| Copenhagen1 | 17 | 1088 | | | |
| Helsinki | 20 | 1280 | | | |
| Paris | 16 | 1024 | | | |
| Warsaw | 15 | 960 | | | |
| Belgrade | 10 | 640 | | | |
| **Total** | **340** | **21,760** | **Total** | **209** | **13,376** |

The remainder of this paper is divided into seven sections. In Section 2, we provide a general introduction to building footprint detection and a brief overview of the state of the art. In Section 3, we describe the methods for BFP extraction used in this study as well as necessary pre- and postprocessing steps. Subsequently, in Sections 4 and 5 we describe in detail the employed satellite image datasets and the experimental setup of our study. Section 6 summarizes the results of our study for all investigated research questions (RQ1-RQ6). Finally, we summarize our findings and outline limitations and future work in Section 7.

## 2. Related Work

Building footprint extraction is essential for population monitoring [4], land cover mapping [11], change detection [12], and disaster management [3]. Extracting building footprints (BFP) manually is a time-consuming and tedious task, which begs for automated extraction methods. Automated BFP extraction is usually performed from aerial images and is a challenging task in itself because of the highly heterogeneous and complex appearance of buildings and the heterogeneity of different development types (rural vs. urban areas or residential vs. industrial areas). Furthermore, different contrast, occlusions (buildings covered by trees, high rise buildings), atmospheric conditions (illuminance, diffuse reflections, fog, cloud cover), and coloring qualities of satellite images complicate the process of building footprint extraction.

There are numerous classification and object detection methods for building footprint extraction in the literature, but many of these methods provide only the coarse estimation of the building footprint (e.g., bounding boxes) [13]. To determine the exact footprint of a building, the boundary must be tracked at the pixel level, for which segmentation methods are a promising means. Many traditional approaches have been investigated for building footprint extraction, such as morphological filtering [14], edge filtering [15], and using texture-based features [16]. Such conventional methods are based on the extraction of image features, i.e., texture, color, and geometric information related to or characteristic of buildings. Approaches building upon such features, however, vary considerably in performance across different lighting conditions and building architectures, leading to class confusions with other land covers, such as roads and parking lots [11]. To cope with these issues and to increase BFP extraction performance, one strategy is to use high-resolution image data, if available [17]. Other methods leverage additional modalities for building

footprint extraction, such as multi-spectral images [14] and LIDAR (light detection and ranging) data [18]. Such data can be extremely helpful but are not available everywhere, which limits the application of such methods to certain regions which have been captured accordingly. The integration of LiDAR technology has certainly impacted the development of methods for BFP extraction. However, the availability of high-resolution LiDAR data for large-scale applications with frequent observation rates is limited exclusively to commercial research due to high acquisition costs [19]. Therefore, the focus of our study is on BFP extraction from RGB data, since this type of data is readily available.

In recent years, the methodology for BFP extraction has shifted strongly towards machine-learning-based approaches, in particular deep learning approaches. This has been fuelled by the increasing availability of large-scale datasets [20] and the great success of convolutional neural networks (CNNs). CNN-based methods are capable of learning high-level image features and thus often perform better than traditional image analysis techniques [21]. Segmentation approaches building upon deep learning are typically based on encoder–decoder architectures or object detection networks [22].

Popular encoder–decoder networks for segmentation are SegNet [9] and U-Net [23]. Both networks have multiple encoder (convolutional) layers and decoder (deconvolution) layers, allow for segmentation at pixel-level, and are frequently used for BFP extraction [24,25]. In [21], a modification of SegNet is proposed to obtain more accurate predictions of pixels at the boundaries of buildings. The network is trained with non-binary masks that emphasize the boundary (obtained via distance transform), which results in a slight performance improvement compared to using only binary labels. Evaluation of the approach is, however, limited to a single, non-publicly available dataset. A multi-task learning approach based on SegNet is proposed in [26] in which distance transform is used as an additional layer in the decoder to overcome the typical over-smoothing of output segments, which is a common artifact in BFP extraction. A similar idea is presented in [21], where the authors not only integrate distance information but further context information from the surrounding of the building to find better boundaries. U-Net is used in [24] to segment buildings from aerial images. The network performs well in detecting buildings but has weaknesses in tracing boundaries where the distance between buildings is small. Results from the literature show that SegNet and U-Net perform at a similar performance level with slight advantages for SegNet, see, e.g., [27,28] who report a performance advantage of around two percentage points for SegNet over U-Net. A particular strength of SegNet seems to be its capability to extract small buildings [21,27] and features corresponding to edges of the building (especially when distance and context information are provided to the network) [21].

Mask R-CNN [10] is a popular object-detection-based segmentation network that is frequently used for building footprint extraction in the literature [29]. Mask R-CNN provides both a bounding box around the building and a pixel-level segmentation mask. In contrast to encoder–decoder architectures, Mask R-CNN further supports instance segmentation, which can help identify individual buildings even if they are attached [30]. The authors in [29] use Mask R-CNN for building extraction and improve the regularity of the predicted building shape in a postprocessing step. Studies on BFP extraction show that Mask R-CNN is a powerful general purpose network that can be adapted to building detection well [31]. There are, however, certain limitations reported in the literature, such as limitations in the detection of small buildings, especially in densely populated built-up regions where buildings are attached to each other [32,33] and for large buildings with complex roof structures (e.g., in industrial areas) [33]. Much better performance for Mask R-CNN is reported for single, free-standing buildings [33]. Furthermore, the performance level of Mask R-CNN stongly depends on the building structure and location, as shown in [29], where strongly differing Dice/F1 scores were obtained depending on the investigated city, e.g., 0.88 for Las Vegas, 0.76 for Paris, 0.65 for Shanghai, and only 0.58 for Khartoum.

More recently, attention-based neural networks have increasingly been used for building segmentation [34–36]. Attention-based networks provide better modeling capabilities for spatial context compared to pure CNNs and thus can improve BFP extraction as shown in [37]. Other approaches build upon multimodal data (e.g., RGB plus LIDAR data) to improve building segmentation [38]. Multiple modalities can be either fused at data-level if they are represented in the same domain (same dimensions) [39] or at model-level, useful when the inputs do not directly align [21]. The fusion of complementary data usually improves results but limits the applicability of the approach to regions where all input modalities are available.

As summarized in [40], many BFP extraction methods involve additional postprocessing steps (e.g., polygonization) to improve building contours, which are based, e.g., on Douglas–Peucker [41,42], mesh approximation [43,44], or pixel approximation considering the entire prediction feature space [45]. The approach presented by [40] uses a fully convolutional network (FCN) for semantic segmentation along with postprocessing for contour refinement using the marching squares algorithm.

In this study, we focus on questions related to the practical applicability of BFP extraction methods. To this end, we selected two complementary architectures which represent the basis for many BFP extraction methods, namely SegNet as representative of a popular encoder–decoder architecture and Mask R-CNN as representative of an object-detection-based segmentation approach. Therefore, we follow the idea of [31] who previously compared semantic segmentation approaches with instance-based segmentation approaches for BFP extraction. The focus of our study is, however, less on achieving peak performance than on evaluating how well the architectures themselves generalize to new (unseen) data, different settlement structures and regions, and how well they can benefit from increased training data and transfer learning with external data. An additional benefit of the study is that we directly compare the two types of network architectures and gain insight into their individual strengths and weaknesses.

## 3. Methodology

### 3.1. Preprocessing

Both publicly available datasets as well as our own dataset (see Section 4), consist of large high-resolution images (e.g., 6000 × 6000 pixel tiles in case of ISPRS and 5000 × 5000 for INRIA). Since the employed network architectures typically require smaller image sizes (patches), we need to perform certain preprocessing operations. Simply downscaling the images would lead to a drastic loss of information; thus, we pursue a patch-based approach. Specifically, we divide each high-resolution input image into multiple overlapping patches and feed each single patch to our segmentation networks. To handle image borders, symmetric auto-completion is used (see Figure 1).

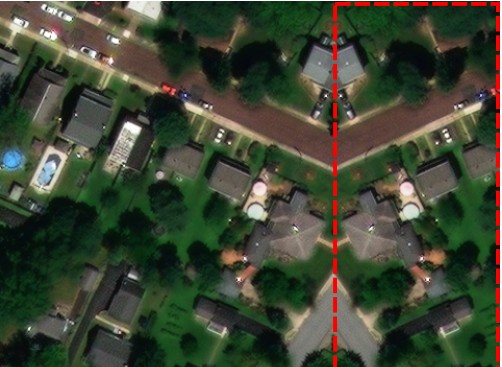 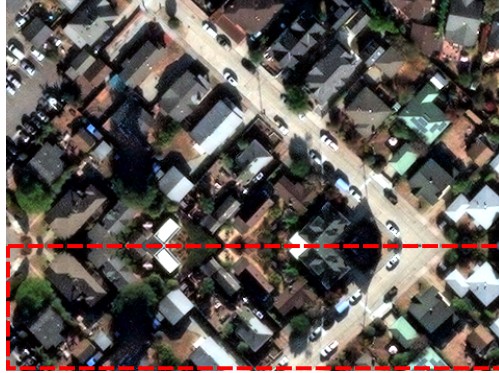

**Figure 1.** Symmetric auto-completion of patches: patches which partly fall outside the domain of the input image are filled with a mirrored version of the image content inside the patch. This avoids unwanted artifacts.

### 3.2. Segmentation Approaches

For our study on BFP extraction, we chose two structurally different network architectures, namely SegNet [9] and Mask R-CNN [10,29], which are described briefly in the following.

**SegNet** is a simple encoder–decoder network architecture (see Figure 2 for an overview). The input of SegNet is an RGB image together with a corresponding ground truth mask with per-pixel class labels. The encoder is composed of 13 convolutional layers, which are directly adapted from the VGG16 architecture [46]. In total, five max-pooling layers are utilized to decrease the feature map sizes throughout the encoder. The decoder also consists of 13 convolutional layers. For upsampling, sparse feature maps are created and activations from the input feature map are placed according to the matching max-pooling position indices of the encoder side. The network is trained end-to-end with cross-entropy loss.

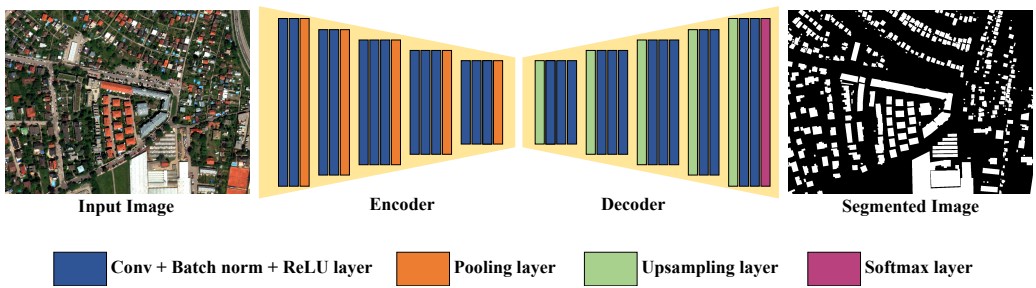

**Figure 2.** The encoder–decoder network architecture of SegNet.

**Mask R-CNN** is a multi-task network that utilizes a two-step procedure to enable instance segmentation (see Figure 3 for an overview of the architecture). Initially, features of incoming RGB images are extracted with a backbone network. For this, various pretrained network architectures can be chosen (e.g., VGG16 or ResNet). In the next step, the computed feature maps are sent to a region proposal network (RPN). The RPN proposes different bounding box position and is based on a sliding window approach with predefined anchors. Subsequently, the bounding box regions are extracted from the computed feature maps and are transformed to a fixed size through a quantization method (ROIAlign). At this stage, the network splits into two separate sub-networks. One branch of the network refines the bounding box parameters and predicts the class of the object within each box. In parallel, in a second branch of the network, a series of convolutional layers is used to predict the segmentation mask of every extracted region of interest (ROI). To train the network, a multi-task loss function is defined which combines losses for segmentation, class segmentation, and bounding box regression.

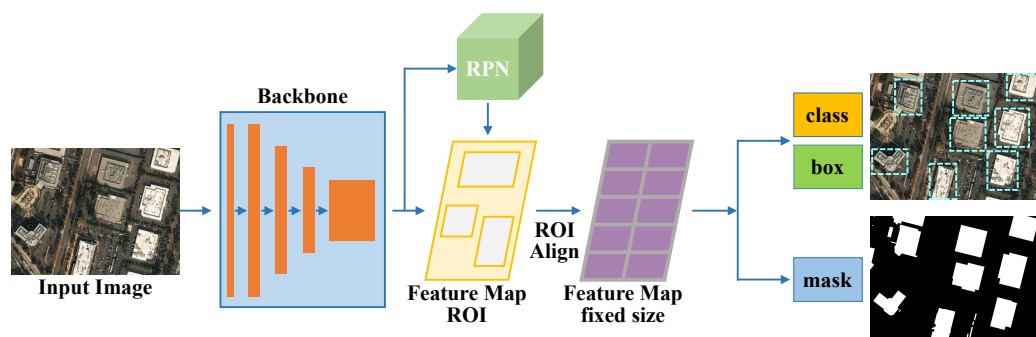

**Figure 3.** The network architecture of Mask R-CNN.

### 3.3. Postprocessing

To obtain a result for the high-resolution input image tiles, the patch-based segmentation outputs from the above networks need to be fused into a large-scale image again. Since we use overlapping patches, in areas of overlap there exist multiple segmentations,

i.e., multiple label predictions for a given pixel. We apply a simple type of voting to decide on a final prediction in which we declare a pixel as belonging to a building footprint if there is at least one positive prediction for this pixel among all overlapping predictions. This strategy facilitates the building class, which has shown to be beneficial as this class is typically underrepresented compared to the background class (usually when taking urban and rural areas into account, the area not covered by buildings is typically larger in average than the area covered by buildings).

## 4. Study Datasets

### 4.1. GDBX Dataset

An evaluation of various data providers showed that Maxar's (formerly DigitalGlobe) (https://www.maxar.com, accessed on 16 February 2018, last visited on 2 December 2022) Geospatial big data platform GBDX from DigitalGlobe is the most suitable for our research task, providing very high resolution (VHR) imagery worldwide with 30 cm resolution. GBDX further offers accessing and processing of Open Street Map (OSM) data, which ease ground truth generation. The dataset compiled from the GDBX platform for our experiments exclusively comprises satellite image tiles from WorldView imagery.

As an initial step, workflows were developed on the GBDX platform for automatically searching for OSM buildings within a defined region of interest (ROI) in Europe and North America. In a second step, cross-matching was applied between OSM and ROI to define three categories of high, medium, and low-density built-up areas in order to obtain a well-distributed sample dataset capturing urban and rural areas. Based on this preanalysis, VHR data were selected for each of the predefined areas. For each of the ROIs, multiple tiles with a height of 1080 px and a width of 1440 px were randomly selected, extracted, and stored as PNG raster files accompanied by the respective WorldView metadata and OSM data.

To avoid unwanted effects, such as large shadowed areas and too large tilt angles (see Figure 4 for examples), we applied filters directly on the GBDX platform that restrict cloud cover to a maximum of 30% and the off-nadir angle to less than or equal to 10 degrees. Exclusively, WorldView-3 and WorldView-4 imagery was retrieved. In cases where multiple tiles were retrieved for the same area, the tile with the lowest off-nadir angle was selected.

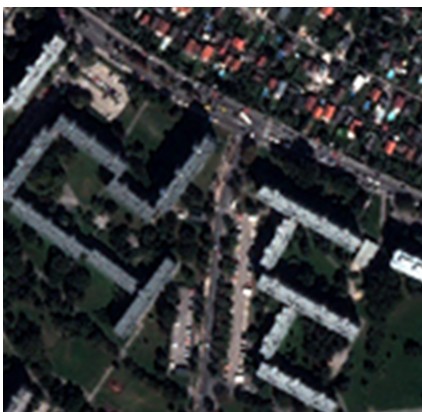 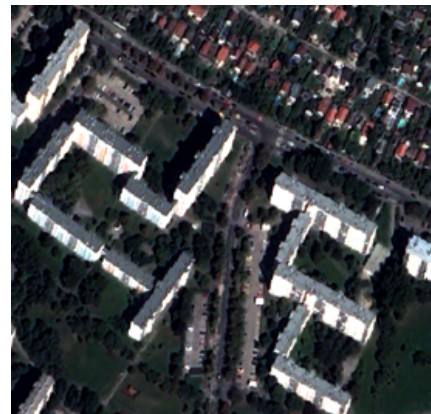

(**a**) Bad sun azimuth angle        (**b**) Bad acquisition angle

**Figure 4.** Unwanted effects in satellite imagery for BFP extraction: (**a**) example of VHR data with suboptimal solar azimuth angle; (**b**) cutout of tile with large off-nadir angle leading to a tilted view.

The compiled image dataset was used in a further step to fetch the respective OSM data for every ROI and to extract the item type "buildings". The outputs were projected from EPSG:4326 (1984 World Geodetic System) into the corresponding UTM zone. Each of the 1080 × 1440 tiles was again verified manually regarding their building density and grouped into low, medium, or high density of buildings. Both the VHR ground truth layer from GDBX and the OSM footprints were downloaded for a subsequent visual interpretation

step. Based on both information sources, ground truths (building footprints) were refined, consolidated, and stored in raster and vector format for 24 cities in Europe and North America. See Table 1 for an overview of the dataset.

### 4.2. INRIA and ISPRS Datasets

ISPRS [8] and INRIA [6] are publicly available datasets suitable for the task of BFP extraction and building segmentation and comprise high-resolution aerial images. Table 2 provides an overview of both datasets in terms of resolution, number of covered cities, tiles, and the tile size and a comparison to the GDBX dataset from Section 4.1.

**Table 2.** Overview of all three datasets used in our study. The GDBX dataset has the largest diversity in terms of different cities. ISPRS stands out in resolution, and INRIA overall has the largest number of patches.

| Dataset | Source | Resolution | No. of Cities | Tile Size (px) | No. of Tiles | No. of Patches |
|---------|--------|------------|---------------|----------------|--------------|----------------|
| GBDX | aerial | 30 cm | 24 | 1080 × 1440 | 549 | 35,136 |
| ISPRS | aerial | 5–9 cm | 2 | 6000 × 6000 | 15 | 45,229 |
| INRIA | aerial | 30 cm | 5 | 5000 × 5000 | 180 | 228,780 |

The ISPRS dataset covers the two cities of Vaihingen and Postdam and provides label information for six different classes (impervious surfaces, building, low vegetation, tree, car and clutter/background). Vaihingen consists of many detached buildings and small multi-story buildings, while Potsdam consists of typical historical buildings with large and dense building blocks and narrow streets. Both cities in ISPRS cover urban scenes.

In comparison, the INRIA dataset provides labels for only two categories (i.e., building and background). The INRIA dataset provides data from both Europe and the United States (US) similar to our dataset, i.e., Austin, Chicago, Kitsap County, Vienna, and West Tyrol (Austria). The INRIA dataset covers densely populated cities as well as less populated cities with different urban settings. Example images for both datasets are shown in Figure 5.

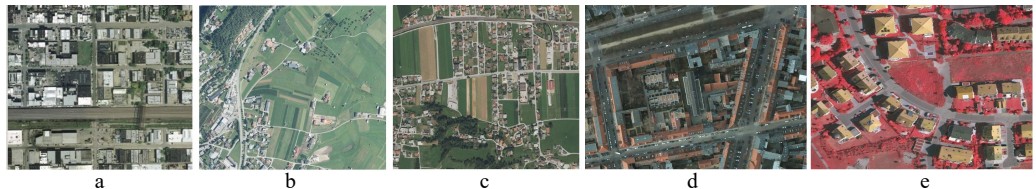

**Figure 5.** Example images from the INRIA dataset (**a**–**c**) and from the ISPRS dataset (**d**,**e**).

## 5. Experimental Setup

### 5.1. Dataset Preparation

We employ all three datasets described in Section 4 in our study. The main dataset is the GBDX dataset. The ISPRS and INRIA datasets are used to evaluate the generalization ability of the employed BFP extraction models in transfer learning settings. For our experiments, we exclusively used RGB information and merged all available class labels such that the only segmentation classes were building and background.

In a preprocessing step, overlapping patches were extracted from the original image tiles with a patch size of 360 × 480. The overlap ratio was set to one-third (horizontal overlap of 120 pixels and vertical overlap was of 160 pixels). The set of tiles in each dataset was split into disjoint training and test sets. Patching was applied only after the dataset was split. The split ratio of training- and test-set was 70%:30%. Since we use different dataset partitions in our experiments, this ratio may vary by a few percentage points between experiments. The input images were used as they were provided in the datasets, i.e., no color enhancement or other type of image enhancement was applied.

### 5.2. Dataset Partitioning

In this study, we want to examine how different dataset characteristics influence the segmentation performance of building footprints and how robust and generalizable the evaluated BFP extraction models are across cities, continents, and different spatial building structure (see research questions RQ1-6 stated in Section 1). To systematically answer these questions, we created different partitions of our datasets, first and foremost the GDBX dataset (see Table 3).

**Table 3.** Overview of the different partitions of the GDBX dataset necessary to answer the individual research questions. GDBX Partition 4 has been performed directly on a patch level, all others on a tile level. Note that for answering RQ3 (transfer learning), we employ the publicly available datasets INRIA and ISPRS (not listed here).

| Partition | Naming | No. of Tiles | No. of Patches |
|---|---|---|---|
| GDBX Partition 1 (RQ1 & RQ2) | $GDBX^S$ | 215 | 13,760 |
| | $GDBX^L$ | 320 | 20,480 |
| GDBX Partition 2 (RQ4) | $GDBX^{EU}$ | 340 | 21,760 |
| | $GDBX^{NAM}$ | 209 | 13,120 |
| GDBX Partition 3 (RQ5) | $GDBX^{SFB}$ | 183 | 11,776 |
| | $GDBX^{MFB}$ | 190 | 12,160 |
| | $GDBX^{IB}$ | 88 | 5632 |
| GDBX Partition 4 (RQ6) | $GDBX^{SDS}$ | - | 10,858 |
| | $GDBX^{BS}$ | - | 8588 |
| | $GDBX^{DDS}$ | - | 6381 |

**GDBX Partition 1 (RQ1 and RQ2)**: The first partition of the dataset serves the investigation of RQ1 and RQ2, i.e., the question if additional training data improves performance and how well the methods generalize to completely unseen cities. To this end, the GDBX dataset (referred to as $GDBX$ in the following) was split into two disjoint datasets: a smaller subset named $GDBX^S$ and a larger one called $GDBX^L$. Together, they form the whole dataset $GDBX$. $GDBX^S$ consists of only ten randomly chosen cities (from both Europe and the US) out of 24 cities, while $GDBX^L$ covers all the other cities.

Both subsets are further split into disjoint training $GDBX^S_{trn}/GDBX^L_{trn}$, validation $GDBX^S_{val}/GDBX^L_{val}$, and test $GDBX^S_{tst}/GDBX^L_{tst}$ sets. Both test sets are further split into a test set containing cities already seen in the training set and a test set with completely unseen cities. We refer to these subsets of the test sets as $GDBX^S_{tst-seen}$ and $GDBX^S_{tst-unseen}$ for the small subset $GDBX^S$ and similarly $GDBX^L_{tst-seen}$ and $GDBX^L_{tst-unseen}$ for the large subset $GDBX^L$. The split between training, validation and test sets is always based on entire tiles.

**GDBX Partition 2 (RQ4)**: The second partition serves the investigation of the generalization ability of BFP extraction across continents (RQ4). To this end, the GBDX dataset is split into European and North American cities, i.e., $GDBX^{EU}$ and $GDBX^{NAM}$. These two subsets are further divided into training, validation, and test sets, i.e., ($GDBX^{EU}_{trn}$, $GDBX^{EU}_{val}$, $GDBX^{EU}_{tst}$ and $GDBX^{NAM}_{trn}$, $GDBX^{NAM}_{val}$, $GDBX^{NAM}_{tst}$). All cities across all partitions are disjoint, i.e., no city appears in more than one subset.

**GDBX Partition 3 (RQ5)**: This partition facilitates the investigation of RQ5, which focuses on the robustness of BFP extraction across different settlement and building structures. For this partition, a subselection of the GDBX data from each city was categorized into: predominantly single family building (SFB), predominantly multiple family buildings (MFB), and predominantly industrial buildings (IB), see Figure 6 for examples. The categorization was made manually by a domain expert judging the predominant type of building in each tile. Each subset is further split into training, test, and validation set, where all

sets are disjoint with respect to the contained cities. The resulting subsets of the data are referred to as: $GDBX_{trn}^{\{SFB|MFB|IB\}}$, $GDBX_{val}^{\{SFB|MFB|IB\}}$, and $GDBX_{tst}^{\{SFB|MFB|IB\}}$.

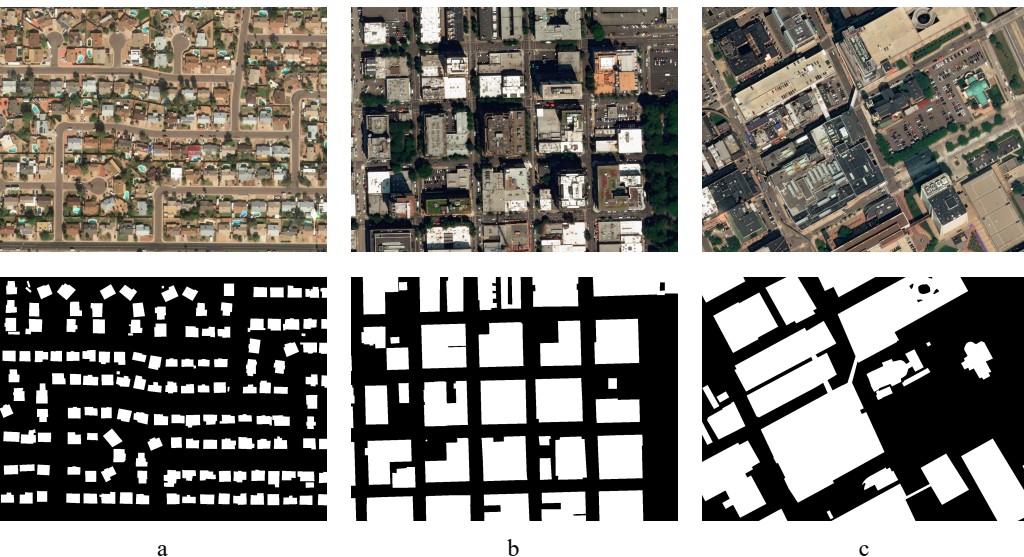

**Figure 6.** Visual example of buildings with different predominant styles (SFB: single family buildings (**a**), MFB: multiple family buildings (**b**) and IB: industrial buildings (**c**)).

**GDBX Partition 4 (RQ6)**: This data partition should enable the investigation of the robustness of BFP extraction for differently dense building structures (RQ6). To this end, the set $GDBX^L$ is divided into three subsets with different building structures/topology, i.e., sparse detached structure (SDS), block structure (BS), and dense detached structure (DDS), see Figure 7 for examples. These three subsets were automatically categorized using a simple heuristic building upon density estimation (number of buildings per area) and the building size (area covered by a building) derived from the ground truth masks. In a first step, we compute the distribution $d_{nb}$ of the number of buildings per patch for all training patches. The heuristic is applied on patch-level, i.e., each patch is categorized into one of the above classes using the following simple rules:

$$SDS : th_{nb} \leq 20 \; and \; th_a \leq 15$$

$$BS : th_{nb} \leq 15 \; and \; th_a \geq 20$$

$$DDS : th_{nb} \geq 25 \; and \; th_a \leq 60,$$

where threshold $th_{nb}$ refers to the number of buildings in a patch (expressed as percentile of the distribution $d_{nb}$ of the number of buildings across all patches), and threshold $th_a$ refers to the area covered by all buildings in one patch (in percent of the total patch area). This means, for example, that patches where (i) the number of buildings is lower than the value corresponding to the 20th percentile of $d_{nb}$ and (ii) the area covered by buildings is higher than 15% of the total patch area are classified as sparse detached structure (SDS). Threshold values between the classes have a certain offset to reduce the fuzzy boundaries between the classes (i.e., to remove patches at the transition between two classes) and to improve the balance of the class priors. The further partitioning of the data into training, validation, and test sets (with seen and unseen cities) for each of the three classes follows the partition of $GDBX^L$ from partition 1.

**INRIA and ISPRS Partition (RQ3)**: To investigate the transfer learning capabilities of the evaluated models (RQ3), we employ two datasets from a different source than GDBX. For our experiments, we use the training and validation split reported in the literature to obtain comparable results. Since no ground truth is available for the test sets, the validation set serves as test set for both datasets.

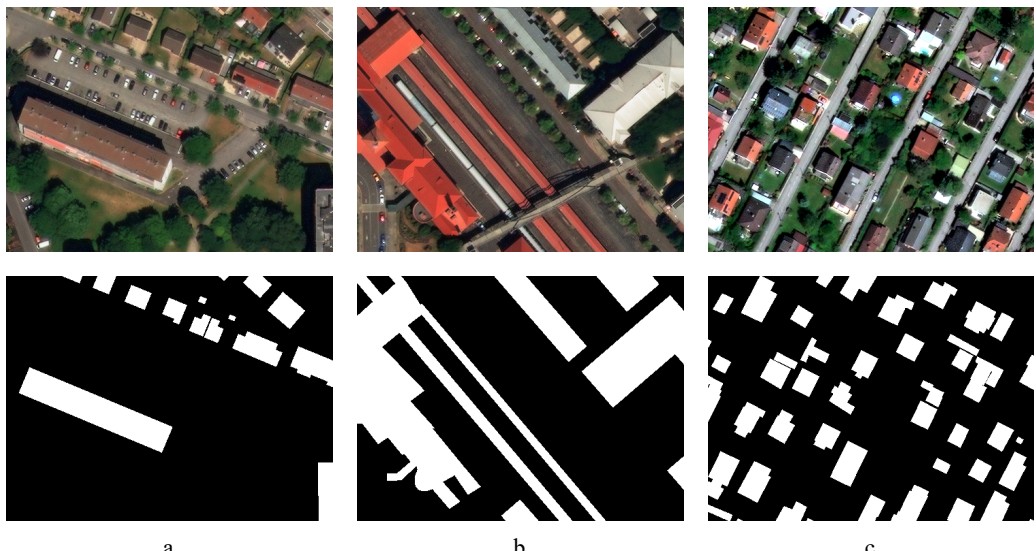

a     b     c

**Figure 7.** Example patches of areas with different densities: sparse detached structure, SDS (**a**), block structure, BS (**b**), and dense detached structure, DDS (**c**), together with their ground truth (second row).

### 5.3. Performance Measures

We employ Intersection over Union (IoU) and the DICE similarity coefficient to assess the performance of our approach which are specifically designed to evaluate segmentation tasks:

$$IoU = \frac{|S \cap GT|}{|S| \cup |GT|} \tag{1}$$

$$DICE = 2\frac{|S \cap GT|}{|S| + |GT|}, \tag{2}$$

where $S$ is the predicted segmentation mask, and $GT$ is the ground truth mask. Note that the DICE similarity coefficient is in the case of a binary segmentation problem equivalent to the F1-score. In addition to IoU and DICE/F1-score, we further assess the performance by accuracy, precision, and recall to obtain a more detailed view on the results and to achieve full comparability with the literature:

$$Accuracy = \frac{TP \times TN}{TP + FP + TN + FN} \tag{3}$$

$$Precision = \frac{TP}{TP + FP} \tag{4}$$

$$Recall = \frac{TP}{TP + FN}, \tag{5}$$

where $TP$, $TN$, $FP$, and $FN$ refer to true positive, true negative, false positive, and false negative pixels.

In our results, we report these performance measures at tile level and averaged over all tiles of the respective tests sets. The only exception is Section 6.7, where we further report results at patch level (averaged over all patches) for comparison.

### 5.4. Network Training

For each experiment, we train both SegNet and Mask R-CNN models. A batch size of five is used for SegNet and a batch size of two for Mask R-CNN (mainly for memory reasons). The size of the input images (i.e., patches generated from the image tiles) is set to $360 \times 480$ pixels for both networks in all experiments. Patch overlap is one-third. Both

models are trained for 50.000 iterations with a learning rate of 0.001. As loss function, we use binary cross-entropy. Since satellite images usually contain more background pixels than building pixels, we apply a class penalty of 5:1 in the loss function, i.e., false classifications of building pixels are penalized five times stronger than false classifications of background pixels. The patch-level segmentations obtained from the segmentation models are fused as described in Section 3.3 to obtain segmentations at tile level.

## 6. Experimental Results

In the following subsections, we present the experiments performed to answer each of the research questions defined in Section 1 and discuss the respective findings.

### 6.1. RQ1: How Does Training Set Size Affect BFP Extraction Performance?

With the following experiments, we want to evaluate how important the size of the training set is for the task of BFP extraction and if the two evaluated network architectures can effectively take benefit from additional training data (RQ1). We principally expect that more data should be beneficial for prediction quality. With our experiments, we want to quantify the degree of improvement that can be achieved. Knowing the effect of additional training data is important in practice as this helps to trade off the costs of preparing additional labeled training data and the expected benefit of it, e.g., if the expected benefit of additional training data is low, it might not justify the costs of generating additional annotations.

To answer RQ1, we conducted different experiments with the first partitioning of the GBDX dataset ($GDBX^S$ and $GDBX^L$), which represents datasets of different sizes. For the experiments, training was performed on different large training sets to see if additional training data improve test performance. The evaluation was performed in parallel on the following four disjoint test sets: $GDBX^S_{tst-seen}$ and $GDBX^S_{tst-unseen}$ as well as $GDBX^L_{tst-seen}$ and $GDBX^L_{tst-unseen}$, i.e., for both the small and the large GDBX partition there is one test set which contains cities that are also included in the respective training set and one test set which exclusively contains cities which are not included in the training sets. Quantitative results are summarized in Table 4. Figure 8 shows qualitative results.

**Table 4.** Results for RQ1, RQ2: performance for test sets with previously seen and unseen cities for SegNet and Mask R-CNN trained on different large training sets, i.e., (a) $GDBX^S_{trn}$ and (b) the full training data $GDBX^S_{trn}$ and $GDBX^L_{trn}$. 'Acc' stands for accuracy, 'Pr' for precision, and 'Re' for recall. The best results are highlighted in bold. This clearly shows that SegNet outperforms Mask R-CNN. The latter architecture can only outperform SegNet in recall (Re), however, at the cost of precision (Pr), leading to overall lower performance (see DICE, which is in the case of binary segmentation equivalent to F1-score).

| | SegNet | | | | | Mask R-CNN | | | | |
|---|---|---|---|---|---|---|---|---|---|---|
| | IoU | DICE | Acc | Pr | Re | IoU | DICE | Acc | Pr | Re |
| **(a) small train set** | | | | | | | | | | |
| $GDBX^S_{tst-seen}$ | **0.56** | **0.69** | **0.87** | **0.81** | 0.62 | 0.46 | 0.61 | 0.78 | 0.49 | **0.84** |
| $GDBX^S_{tst-unseen}$ | **0.52** | **0.67** | **0.92** | **0.85** | 0.59 | 0.43 | 0.59 | 0.82 | 0.45 | **0.86** |
| $GDBX^L_{tst-seen}$ | **0.55** | **0.70** | **0.89** | **0.83** | 0.63 | 0.49 | 0.65 | 0.82 | 0.54 | **0.87** |
| $GDBX^L_{tst-unseen}$ | **0.55** | **0.70** | **0.87** | **0.82** | 0.63 | 0.47 | 0.62 | 0.75 | 0.52 | **0.86** |
| **(b) full train set** | | | | | | | | | | |
| $GDBX^S_{tst-seen}$ | **0.63** | **0.75** | **0.94** | **0.79** | 0.77 | 0.55 | 0.68 | 0.90 | 0.69 | **0.78** |
| $GDBX^S_{tst-unseen}$ | **0.64** | **0.78** | **0.94** | **0.80** | 0.77 | 0.52 | 0.68 | 0.89 | 0.61 | **0.80** |
| $GDBX^L_{tst-seen}$ | **0.65** | **0.77** | **0.91** | 0.58 | **0.82** | 0.58 | 0.72 | 0.89 | **0.71** | 0.77 |
| $GDBX^L_{tst-unseen}$ | **0.61** | **0.75** | **0.87** | **0.73** | **0.80** | 0.55 | 0.70 | 0.85 | 0.67 | 0.77 |

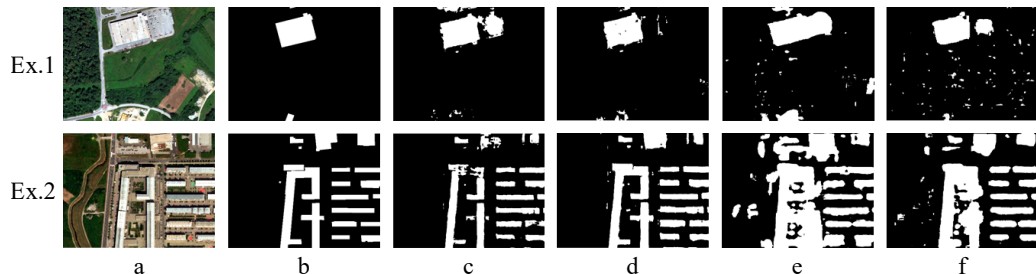

**Figure 8.** Impact of training set size (RQ1): two examples (Ex. 1, Ex. 2) from $GDBX^S_{tst-unseen}$ (small test set with unseen cities) and the predicted segmentation masks obtained by our models: (**a**) input image; (**b**) ground truth; (**c**) SegNet trained on a small training set; (**d**) SegNet trained on full training data; (**e**) Mask R-CNN trained on a small training set; (**f**) Mask R-CNN trained on full training data.

To investigate the impact of the dataset size on BFP extraction performance, we first train both SegNet and Mask R-CNN on the smaller training set $GDBX^S_{trn}$ (see Table 4 (a)) and then repeat the training process with the much larger combined training set consisting of $GDBX^S_{trn}$ and $GDBX^L_{trn}$ (see Table 4 (b)). The results show that for both models, *the performance across all test sets increases when a larger dataset is used for training*. The DICE scores, for instance, show improvements of 5 to 12 percentage points. This positive trend is also reflected in the qualitative results shown in Figure 8. An increase of training set size yields fewer misclassifications (e.g., parking lot in Figure 8c,d, top row) and more accurate object borders (Figure 8c–f, bottom row). These results are in line with previous results on alternative methods in the literature. For example, Refs. [47,48] demonstrate that increasing training set size improves the performance of traditional machine learning techniques (e.g., SVM, random forests) that build upon handcrafted features for remote sensing. In [28], the effect of training set size is investigated for two encoder–decoder models (SegNet and U-Net) for the related task of land cover segmentation, showing that increasing training data facilitates performance. Note that here the SegNet architecture even outperforms U-Net.

Furthermore, we observe that the performance level achieved on the smaller test sets from $GDBX^S$ can be kept and even outperformed by both network models when evaluated on the larger test sets from $GDBX^L$. This indicates a *good generalization ability of both networks*, i.e., more diversity in the test set does not yield a performance drop.

Aside from the general performance increase, the experiment reveals an interesting pattern: for SegNet, increasing training set size primarily facilitates recall while precision even slightly drops. For Mask R-CNN, increasing training set size primarily facilitates precision, while recall drops slightly. This shows that Mask R-CNN detects buildings well, even with little training data and mainly suffers from false positive detections. SegNet, on the contrary, seems to be more robust to false positive detections but misses many buildings (false negatives) when the training set is small. This is against our expectation that more training data should improve both recall and precision. It further indicates that both networks have individual and complementary strengths and weaknesses which—in theory at least—would make the two approaches promising candidates for combination.

When directly comparing the two network architectures, *SegNet outperforms Mask R-CNN in all side-by-side comparisons* (see Table 4). This is also indicated by the visual results in Figure 8. In particular, Mask R-CNN fails to recover the details of complex building structures such as the one on the left side of the bottom image of Figure 8f. One reason for the lower performance of Mask R-CNN might be the additional pooling step that is used to resize all bounding box feature maps to the same size. This means that large object candidates are downsampled to a higher degree. This results in an information loss and may be the reason for less precise segmentation borders.

*6.2. RQ2: How Well Do the BFP Extraction Networks Generalize to Previously Unseen Cities?*

Next, we investigate the robustness of BFP extraction to previously unseen cities, i.e., to new data that was not present during training (RQ2). These findings provide us insights on (i) the robustness of the models and (ii) the necessary heterogeneity that is needed in the training set to obtain results that generalize well. Quantitative results are summarized in Table 4 in the previous section.

In our training and test sets, images of cities from both Europe and North America are included. For example, images from San Jose and Brussels are included in both the $GDBX^S_{trn}$ training set and the $GDBX^S_{tst-seen}$ test set. The test set with unseen cities $GDBX^S_{tst-unseen}$, however, contains images from Detroit and Zagreb, two cities which are not included in the training set. The same applies to the test and train partitions of $GDBX^L$. All training and test sets are balanced across continents (Europe and North America) in the number of contained cities.

The evaluation of BFP extraction techniques on previously unseen cities is important to test generalization ability and has been performed previously in the literature. Several benchmark datasets exist, such as the INRIA dataset where the test set comprises images from cities, which are not included in the training and validation set. Robust BFP extraction methods should generalize well to previously unseen cities. This is, however, not always the case as, for example, shown in [49], where the authors report larger performance drops for unseen cities. Thus, this question is relevant, especially since we investigate a new combination of dataset and segmentation models in this study. Note that in addition to most previous works, we also evaluate our models on a test set comprising images from previously seen cities, which cover a different location than the images in the training set. Therefore, we can reach a better understanding on how attuned the models are to a distinct city structure.

Results of our experiments in Table 4 show that the performance achieved on the test sets containing unseen cities is only marginally lower than for the test sets with previously seen cities. Moreover, we can observe that DICE scores are slightly lower for unseen cities, however only with a maximum of three percentage points. In one case, there is even no performance drop (Mask R-CNN on $GDBX^S_{tst-seen}$ vs. $GDBX^S_{tst-unseen}$). From these results, *we can conclude that both networks have a strong generalization ability across unseen cities. This result is different from the results reported in [49] and speaks for the two investigated network architectures.* Another reason for the good results might be the fact that both European and North American cities are included in the training sets, yielding the necessary diversity during training. We investigate the robustness across cities from different continents further in Section 6.3.

The overall performance level of the two architectures is, however, quite different in these experiments, i.e., average DICE of 0.73 for SegNet vs. 0.65 for Mask R-CNN for experiments on unseen cities. *We thus conclude that SegNet generally outperforms Mask R-CNN in the experiments on RQ2. The same actually applies to the results on RQ1.* Interestingly, these results are complementary to results previously reported in the literature, where Mask R-CNN clearly outperformed encoder–decoder-based networks, such as U-Net, in a study on the Waterloo dataset [31].

*6.3. RQ3: Does Transfer Learning Help in BFP Extraction?*

The three datasets (GBDX, INRIA, ISPRS) that we employ in our study are very different in terms of resolution and recorded areas (see Table 2). Using multiple data sources for training could potentially help the BFP extractor to generalize the concept of building footprints and thus increase the overall accuracy. The central questions here are, (i) how well the investigated network architectures can transfer their knowledge learned from one dataset to another dataset and (ii) if pretraining on one dataset and fine-tuning on another dataset overall increases performance compared to training only on one dataset.

Transfer learning in the context of the segmentation of land covers and building footprints has been previously applied and evaluated in the literature, but either not on

the two investigated models or in a different way. In most cases, transfer learning refers to building upon models that were pretrained on either large image classification datasets, such as ImageNet [50–53], or on segmentation datasets from other domains, such as MS COCO [54,55]. Here, we investigate the effects of transfer learning from one satellite dataset to another one, which has to the best of our knowledge not been investigated for the models employed in this study before.

To answer the above questions, we performed transfer learning experiments again with both SegNet and Mask R-CNN. We conducted experiments for three different scenarios. First, we trained our models on each of the training sets of INRIA and ISPRS separately and evaluated them on the respective test sets of the same dataset to determine the baseline performance (column a in Table 5). Next, we trained the networks only on the GBDX dataset and evaluated them on the test sets of INRIA and ISPRS to estimate how well the networks can generalize from one dataset to the other (column b in Table 5). Finally, we evaluated the efficacy of transfer learning by first pretraining both models on the GBDX training set and then fine-tuning the networks on the training sets of INRIA and ISPRS, respectively (column c in Table 5). The results are summarized in Table 5 and Figure 9. The metric shown in Table 5 is the DICE score achieved on the test sets of INRIA and ISPRS, respectively.

**Table 5.** Impact of transfer learning (RQ3), i.e., pretraining and fine-tuning. DICE scores of SegNet and Mask R-CNN for the test sets of INRIA/ISPRS when trained on (a) INRIA/ISPRS, (b) GBDX, and (c) pretrained on GBDX and fine-tuned on INRIA/ISPRS.

| | SegNet | | | | Mask R-CNN | | |
|---|---|---|---|---|---|---|---|
| | **(a) Self** | **(b) GBDX** | **(c) Finetuning** | | **(a) Self** | **(b) GBDX** | **(c) Finetuning** |
| INRIA | 0.61 | 0.43 | 0.63 | INRIA | 0.48 | 0.56 | 0.44 |
| ISPRS | 0.84 | 0.43 | 0.73 | ISPRS | 0.49 | 0.62 | 0.71 |

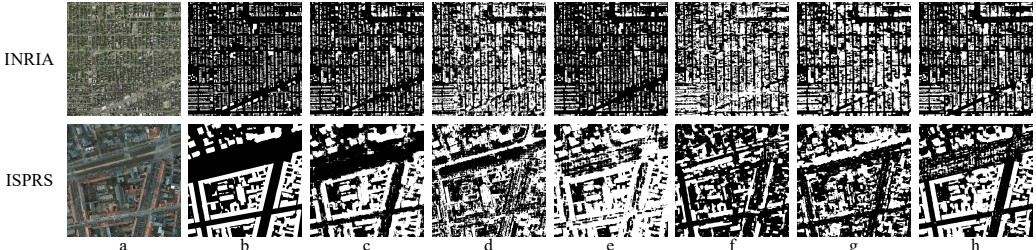

**Figure 9.** Impact of pretraining and fine-tuning (RQ3): two images from the INRIA (top row) and ISPRS (bottom row) test sets and the predicted masks by our models: (**a**) input image, (**b**) ground truth, (**c**) SegNet trained on INRIA/ISPRS, respectively; (**d**) SegNet trained on GBDX only; (**e**) SegNet pretrained on GBDX and fine-tuned on INRIA/ISPRS; (**f**) Mask R-CNN trained on INRIA/ISPRS, (**g**) Mask R-CNN trained on GBDX only, (**h**) Mask R-CNN pretrained on GBDX and fine-tuned on INRIA/ISPRS.

For the experiments with SegNet and INRIA, we can observe that pretraining with GDBX and then fine-tuning helps to increase the DICE score by two percentage points. This is, however, not the case for the ISPRS dataset. Here, pretraining and fine-tuning leads to DICE scores lower than the baseline in column (a). We assume that this varying behavior is due to the differences in resolution. ISPRS has 5–9 cm resolution, and GBDX and INRIA both have 30 cm resolution and are thus more similar. The difference in resolution between ISPRS and GDBX seems to impede transfer learning for the SegNet architecture.

For Mask R-CNN, we observe a different trend. For both datasets, training with GDBX only is highly beneficial. The difference in resolutions seems to be negligible for the Mask R-CNN architecture, which might be due to the multi-scale nature (pooling) of the integrated object detector. For ISPRS, a further increase of performance can be observed

when pretraining with GDBX and fine-tuning with ISPRS data. Surprisingly, we cannot see the same behavior for INRIA, where fine-tuning is not successful.

Overall, we conclude that *there is no clear answer to RQ3*. In three of four cases, pretraining and fine-tuning (c) improve results over only pretraining (b), indicating that transfer learning is beneficial. Compared to the baselines (a), however, the transfer learning results (c) are better in only two of four cases. The effect of transfer learning seems to strongly depend on the employed network model and the characteristics of the dataset. Again, we observe that *SegNet reaches a higher performance level than Mask R-CNN*, see, e.g., the first column labeled "self" in Table 5.

*6.4. RQ4: How Well Do the BFP Extraction Networks Generalize to Cities from Different Continents?*

For RQ2, we already investigated the robustness of the trained BFP extractors to unseen cities, which were not included in the training set. Here we extend the analysis on unseen data to images that were taken on another continent (RQ4). Specifically, we evaluate how models trained on only data from one continent (Europe/North America) perform on data from the other continent.

To this end, we employ the GDBX partition 2 (see Table 3), i.e., the two datasets $GDBX^{EU}$ and $GDBX^{NAM}$, which consist only of cities taken from Europe and North America, respectively. Again, we train SegNet and Mask R-CNN on both datasets, resulting in a total of 4 training sessions. After each session we evaluate the DICE scores for both $GDBX^{EU}$ and $GDBX^{NAM}$ test sets. Results are summarized in Table 6. Example results can be seen in Figure 10.

**Table 6.** Generalization across continents (RQ4): DICE scores of SegNet and Mask R-CNN trained on European data (EU) and North American data (NAM) and tested for test sets of both continents.

|  | **SegNet** | | |  | **Mask R-CNN** | |
|  | **Test: NAM** | **Test: EU** |  | **Test: NAM** | **Test: EU** |
| --- | --- | --- | --- | --- | --- |
| Train: NAM | 0.62 | 0.66 | Train: NAM | 0.71 | 0.67 |
| Train: EU | 0.71 | 0.73 | Train: EU | 0.70 | 0.69 |

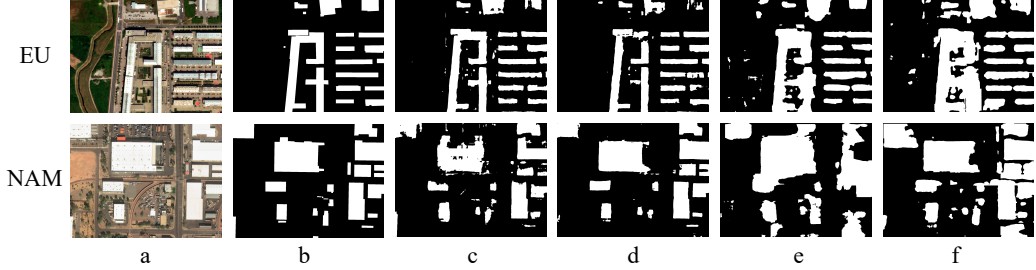

**Figure 10.** Generalization across continents (RQ4): two images from the test sets from Europe (EU) and North America (NAM) together with the predicted masks by our models: (**a**) input image; (**b**) ground truth; (**c**) SegNet trained on EU; (**d**) SegNet trained on NAM; (**e**) Mask R-CNN trained on EU; (**f**) Mask R-CNN trained on NAM.

With SegNet, higher DICE scores for both test sets are reached when training is performed with $GDBX^{EU}$. This is most likely due to fact that the $GDBX^{EU}$ training set (12,864 patches) is much larger than the $GDBX^{NAM}$ training set (6336 patches). Interestingly, SegNet trained on North America ($GDBX^{NAM}$) performs better on the European test set ($GDBX^{EU}$) than on the North American test set ($GDBX^{NAM}$). When trained on $GDBX^{EU}$, the difference between both test set DICE scores is only two percentage points.

When training and testing on North American data with Mask R-CNN, higher DICE scores than with SegNet are achieved. The reason for this could lie in the architecture of Mask R-CNN. Mask R-CNN first extracts bounding boxes and then performs segmentation for the detected objects. The often observed box-like industrial building structures in North American cities and the suburbs full of standalone one-family houses are therefore very

convenient for the architecture of Mask R-CNN. This could further explain why Mask R-CNN is outperformed by SegNet for European data. SegNet may cope better with the more complex building shapes and attached buildings in European cities, yielding a higher performance (0.73 for SegNet vs 0.69 for Mask R-CNN).

Concerning the generalization ability across continents, we can conclude that *both SegNet and Mask R-CNN generalize well to data from the other continent*. Therefore, the generalization ability is higher when training is performed on the European cities. This may be due to two reasons, first the European data partition is larger (see above), and second, the heterogeneity in building structures in this training set is larger. Overall, we observe that the performance level of the two models (SegNet and Mask R-CNN) is comparable in these experiments.

### 6.5. RQ5: Is BFP Extraction Robust across Different Settlement/Building Structures?

Different districts and areas of a city may have a characteristic building type, e.g., industrial area with large factory and storage buildings vs. a suburb with many similar looking single family houses. Here, we want to investigate how a bias towards a particular building type in the training set affects the overall BFP extraction performance.

To this end, we extract three subsets from the GDBX data containing different dominant building structures (see also GDBX partition 3 in Section 5.2), i.e., single family buildings ($GDBX^{SFB}$), multiple family buildings ($GDBX^{MFB}$), and industrial buildings ($GDBX^{IB}$). Both networks (SegNet and Mask R-CNN) are trained separately on training splits from these three subsets and evaluated on respective test partitions. All combinations of building structures in test and training splits are evaluated, yielding nine experiments per network architecture. For evaluation, we compute the DICE scores for each training/test combination, see Table 7. Figure 11 shows the qualitative results for the experiments.

**Table 7.** Robustness to different building structures (RQ5). Test DICE scores of SegNet and Mask R-CNN trained on single family buildings (SFB), multiple family buildings (MFB), and industrial buildings (IB), respectively.

| | SegNet | | | | Mask R-CNN | | |
| | Test: SFB | Test: MFB | Test: IB | | Test: SFB | Test: MFB | Test: IB |
|---|---|---|---|---|---|---|---|
| Train: SFB | 0.70 | 0.68 | 0.59 | Train: SFB | 0.47 | 0.54 | 0.46 |
| Train: MFB | 0.70 | 0.74 | 0.52 | Train: MFB | 0.47 | 0.55 | 0.52 |
| Train: IB | 0.64 | 0.71 | 0.73 | Train: IB | 0.33 | 0.41 | 0.39 |

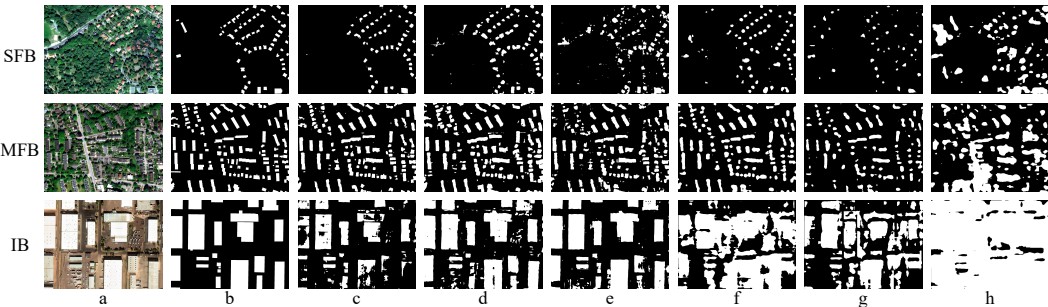

**Figure 11.** Robustness to different building structures (RQ5): three image examples from the test sets with single family buildings (SFB), multiple family buildings (MFB), and industrial buildings (IB) together with the predicted masks by the models: (**a**) input image; (**b**) ground truth; (**c**) SegNet trained on SFB; (**d**) SegNet trained on MFB; (**e**) SegNet trained on IB; (**f**) Mask R-CNN trained on SFB; (**g**) Mask R-CNN trained on MFB; (**h**) Mask R-CNN trained on IB.

The results for SegNet show that training and evaluating on the same building structure (diagonal entries in the Table 7) yield the best results. When training and testing are performed on different building types, performance typically drops. This drop is relatively

small between SFB and MFB (zero to six percentage points). For IB, the drop is much larger (6 to 21 percentage points), i.e., SegNet trained on IB works well when testing is performed on the same building type but generalizes poorly for SFB and MFB. Figure 11 gives an indication of why this may be the case. SFB and MFB houses have more similar building sizes compared to IB, which are much larger and partially denser. This impedes generalization from IB to SFB and MFB.

In comparison to SegNet, we observe much lower DICE scores in the experiments with Mask R-CNN. Given that such a large performance difference did not exist in the previous experiments, we assume that the size of the training sets is the main reason for this behavior. Since the data partitions $GDBX^{SFB}$, $GDBX^{MFB}$, and $GDBX^{IB}$ contain only those tiles from the GDBX dataset where one dominant building type could be clearly identified, these data partitions are smaller than in previous experiments and less data are available for training (e.g., $GDBX^{EU}$ used in Section 6.4 comprises 12864 training images, while $GDBX^{IB}$ contains only 2880 images).

Overall, we can clearly see that building structure has a strong impact on the overall performance. *The generalization to different building structures is limited and especially low when the sizes of the buildings differ significantly.* Furthermore, we observe that Mask R-CNN requires more training data than SegNet to achieve a comparable performance level. In case of too little training data (e.g., in case of IB), the network fails completely (DICE score of 0.39). The effects of insufficient training data in the case of Mask R-CNN can also clearly be seen in the predictions shown in Figure 11f–h. Difficulties of Mask R-CNN with large buildings (as in the case of IB) have also been reported in the literature in previous experiments [33]. Our results support these observations.

### 6.6. RQ6: How Well Do the BFP Extraction Networks Generalize to Differently Dense Areas?

In addition to the building type, different regions can also vary in the overall density of the constructed buildings. This aspect is especially relevant to BFP extraction, since correctly segmenting building boundaries poses a more difficult challenge if buildings are connected or very close to one another.

Here, we evaluate the performance and robustness of BFP extraction for three different spatial building structures, i.e., sparse detached structure ($GDBX^{SDS}$), block structure ($GDBX^{BS}$), and dense detached structure ($GDBX^{DDS}$). Again, we conduct experiments with both network architectures and evaluate all combinations of training and test data. Additionally, we split the test data into a test set with previously seen cities and completely unseen cities (as in Section 6.2). The resulting DICE scores for the test sets can be found in Table 8 (previously seen cities) and Table 9 (previously unseen cities). Qualitative examples are shown in Figure 12.

In the results, we observe a similar pattern to the previous experiment in Section 6.5. First, SegNet outperforms Mask R-CNN in all experiments by a large margin. The most likely reason is again the smaller sized training sets. Second, similar to the experiments in Section 6.5, generalization across different building types/densities is better for more similar classes (here SDS and DDS) and worse for structurally different types (here, in particular BS). Models trained on DDS also yield good results for SDS and vice versa. Models trained on BS, however, fail to model SDS and DDS well. This is the case for both test sets, i.e., with seen cities and unseen cities. Figure 12 provides example images for the different density classes and clearly shows their different structure. DDS and SDS buildings are mostly similar in size but different in spatial density which facilitates generalization. More challenging for the networks is to generalize from large to small building structures (which is consistent with our observations from the experiments in Section 6.5), i.e., from BS to SDS and DDS, where BS is very different in terms of building size and geometry. This also becomes evident when looking at Tables 8 and 9. SegNet trained on BS achieves peak performance only for BS. For SDS and DDS, there is a notable performance drop, which shows that generalization works poorly in this case. The effect is smaller for Mask R-CNN and partly masked by the generally poorer performance for this network in this experiment.

Comparing test results for seen cities (Table 8) and unseen cities (Table 9), we find that in 16 out of 18 total cases, the model performance on the test set with seen cities is equal or higher than for unseen cities. This is according to our expectations that unseen cities pose an additional challenge to the networks with respect to generalization ability. Nevertheless, the performance difference on average is small. For SegNet, the average performance in all experiments on seen cities is an average DICE score of 0.68. For unseen cities, this average score drops to 0.66. For Mask R-CNN, the performance drop on average is 4 percentage points (from 0.45 to 0.41 on average), however, ata much lower performance level. From these experiments, *we can conclude that generalization to unseen cities works better than generalization to different building densities and building structures*. Heterogeneity in building/settlement structures is thus particularly important in the training set.

**Table 8.** Robustness to different building density (RQ6). Test set DICE scores of SegNet and Mask R-CNN trained on areas with sparse detached structure (SDS), block structure (BS), and dense detached structure (DDS), respectively. The test data contain cities already seen during training.

| | SegNet (Seen Cities) | | | | Mask R-CNN (Seen Cities) | | |
|---|---|---|---|---|---|---|---|
| | Test: SDS | Test: BS | Test: DDS | | Test: SDS | Test: BS | Test: DDS |
| Train: SDS | 0.64 | 0.61 | 0.70 | Train: SDS | 0.32 | 0.30 | 0.34 |
| Train: BS | 0.55 | 0.81 | 0.65 | Train: BS | 0.27 | 0.59 | 0.45 |
| Train: DDS | 0.66 | 0.72 | 0.78 | Train: DDS | 0.50 | 0.62 | 0.63 |

**Table 9.** Robustness to different building density (RQ6): test set DICE scores of SegNet and Mask R-CNN trained on areas with sparse detached structure (SDS), block structure (BS), and dense detached structure (DDS), respectively. The test data contain cities not seen during training.

| | SegNet (Unseen Cities) | | | | Mask R-CNN (Unseen Cities) | | |
|---|---|---|---|---|---|---|---|
| | Test: SDS | Test: BS | Test: DDS | | Test: SDS | Test: BS | Test: DDS |
| Train: SDS | 0.60 | 0.55 | 0.73 | Train: SDS | 0.30 | 0.28 | 0.33 |
| Train: BS | 0.52 | 0.77 | 0.66 | Train: BS | 0.22 | 0.56 | 0.39 |
| Train: DDS | 0.63 | 0.68 | 0.78 | Train: DDS | 0.46 | 0.58 | 0.61 |

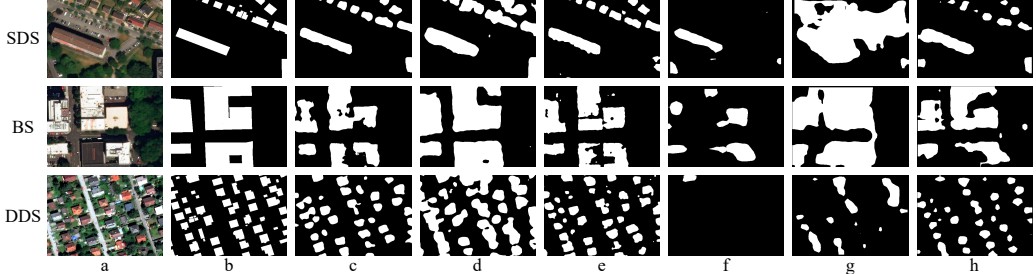

**Figure 12.** Robustness to different building density (RQ6): three example images from the test sets of $GDBX^{SDS}$, $GDBX^{BS}$, and $GDBX^{DDS}$, together with the predicted masks by the models: (**a**) input image; (**b**) ground truth; (**c**) SegNet trained on SDS; (**d**) SegNet trained on BS; (**e**) SegNet trained on DDS; (**f**) Mask R-CNN trained on SDS; (**g**) Mask R-CNN trained on BS; (**h**) Mask R-CNN trained on DDS.

### 6.7. Additional Aspects

#### 6.7.1. Patching

As described in Section 3.1, we train our models on overlapping patches (of 360 × 480 pixels) extracted from the image tiles (e.g., in case of GBDX of 1080 × 1440 pixels). For performance evaluation, we fuse the patch-level predictions to obtain building footprints at tile level (see Section 3.3). Performance in our experiments is thus reported on the tile level. The question is how much influence the patch fusion has on the overall results. With the following experiment, we want to investigate whether patch-level fusion has a positive

effect on performance. For this purpose, we take SegNet trained on $GDBX^S$ and $GDBX^L$ (compare Table 4 (b)) and evaluate it on the tile and patch level. The results are summarized in Table 10.

**Table 10.** Performance comparison of tile-level evaluation (after patch-level fusion) and patch-level evaluation (without fusion).

| Data Set | Tile Level | | | | | Patch Level | | | | |
|---|---|---|---|---|---|---|---|---|---|---|
| | IoU | DICE | Acc | Pr | Re | IoU | DICE | Acc | Pr | Re |
| $GDBX^S_{tst-seen}$ | 0.63 | 0.75 | 0.94 | 0.79 | 0.77 | 0.64 | 0.75 | 0.94 | 0.80 | 0.78 |
| $GDBX^S_{tst-unseen}$ | 0.64 | 0.78 | 0.94 | 0.80 | 0.77 | 0.59 | 0.71 | 0.93 | 0.80 | 0.71 |
| $GDBX^L_{tst-seen}$ | 0.65 | 0.77 | 0.91 | 0.58 | 0.82 | 0.62 | 0.73 | 0.92 | 0.76 | 0.79 |
| $GDBX^L_{tst-unseen}$ | 0.61 | 0.75 | 0.87 | 0.73 | 0.80 | 0.59 | 0.71 | 0.90 | 0.73 | 0.78 |

On average, the BFP extraction performance in terms of IoU and DICE is better at the tile level than on the patch level (average DICE of 0.76 at the tile level vs. average DICE of 0.73 at the patch level). This demonstrates the beneficial effect of fusing redundant predictions in overlapping parts of the patches. One reason for this beneficial effect is as follows. At the patch level, buildings at the patch border might be cut off and thus not detected. Patch-level fusion can cope with this situation well because due to patch overlap, at the patch boundary at least one adjacent patch for each building exists, where the building is fully visible and thus can be properly segmented.

6.7.2. Sensitivity

An important question in light of the reported results in this study is how robust the BFP extraction is with respect to the selected hyperparameters, in particular the decision threshold that finally decides if a pixel is assigned to the building or the background class. To analyze the dependency of the segmentation performance on the decision threshold, we take SegNet as the more promising network architecture in our experiments and evaluate its performance on different values of the decision threshold (ranging from 0 to 1). From the resulting performance estimates, we compute the receiver-operating (RO) curve and the precision-recall (PR). To maximize the informative value of the sensitivity analysis, we choose the largest test sets in our study, i.e., $GDBX^L_{tst-seen}$ and $GDBX^L_{tst-unseen}$ for this analysis. Figure 13 shows the respective RO and PR curves for the two test sets, together with their area under the curve (AUC) values, i.e., ROC-AUC and pr-AUC. The RO and PR curves show the trade-off between true positives and false positives as well as precision and recall, respectively, which are usually inversely related to each other. Since our BFP extraction is typically an imbalanced segmentation problem (i.e., there are more background pixels than foreground pixels), the PR curve is the more appropriate (and stricter) performance indicator and best shows the optimal working point of the approach.

To investigate in more detail how dependent the segmentation performance is on the decision threshold, we plot the individual performance metrics over the threshold value directly in Figure 14 for both test sets. The plot reveals that there is a quite broad range of decision threshold values (i.e. approx. 0.3–0.6), where the overall performance in terms of DICE/F1 and IOU is kept at a high level. This shows a robust behavior of the approach with respect to the selection of the decision threshold. We can further clearly see the inverse relation between recall and precision as well as a strong correlation between DICE/F1 and IOU.

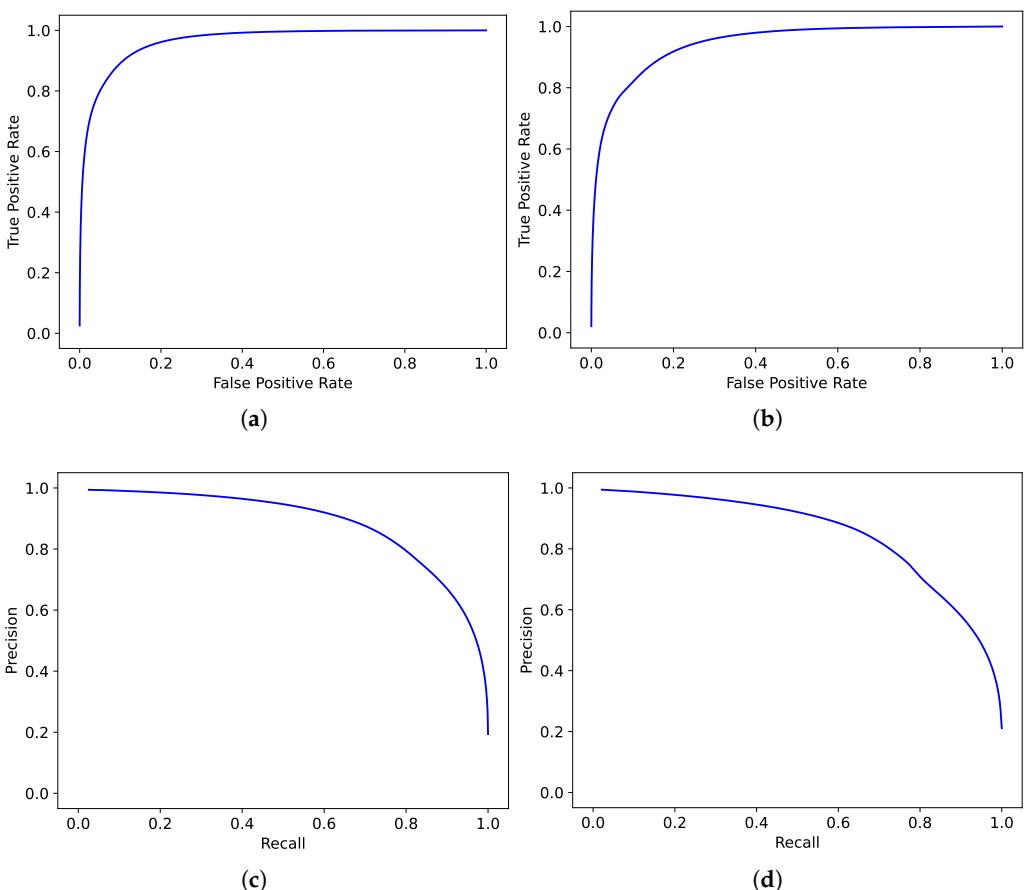

**Figure 13.** Receiver operating and precision-recall curves for the two largest test sets in our study: $GDBX^L_{tst-seen}$ and $GDBX^L_{tst-unseen}$ together with the respective areas under the curve (AUC) values: (**a**) receiver operating curve (ROC) for $GDBX^L_{tst-seen}$, AUC = 0.964, (**b**) receiver operating curve (ROC) for $GDBX^L_{tst-unseen}$, AUC = 0.944; (**c**) precision-recall curve (PRC) for $GDBX^L_{tst-seen}$, AUC = 0.872; (**d**) precision-recall curve (PRC) for $GDBX^L_{tst-unseen}$, AUC = 0.838.

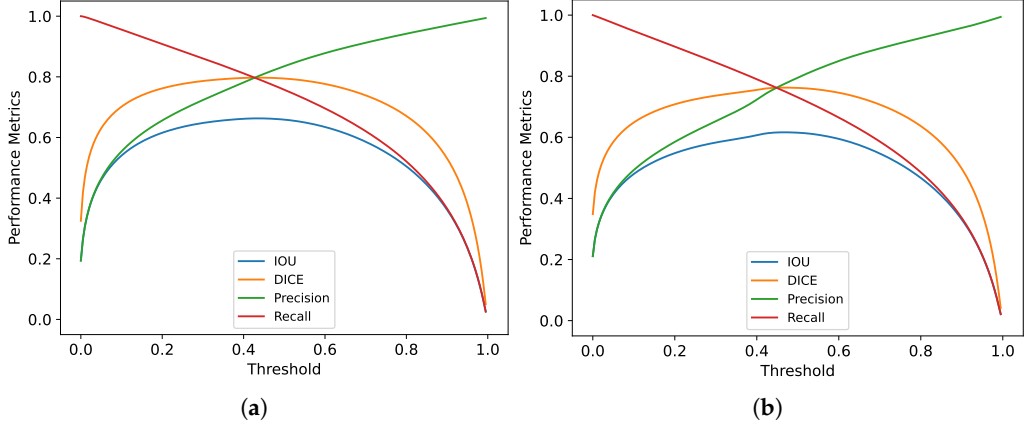

**Figure 14.** Sensitivity of the individual performance metrics on the actual value of the decision threshold. The optimal decision threshold with respect to DICE (=F1) score is 0.44 for $GDBX^L_{tst-seen}$ and 0.47 for $GDBX^L_{tst-unseen}$: (**a**) threshold sensitivity for $GDBX^L_{tst-seen}$; (**b**) threshold sensitivity for $GDBX^L_{tst-unseen}$.

### 6.7.3. Runtime

Our experiments have shown different strengths and weaknesses for the two investigated network architectures. For practical application, another important criterion is the time efficiency of the networks in training and during inference. We have compared the training time of both networks and come to the following conclusion. The training of SegNet is approx. 2.4 times faster than that of Mask R-CNN, which clearly shows the advantage of a one-stage model over a two-stage model. The performance difference during inference is, however, inverse and larger. Here, Mask R-CNN outperforms SegNet, i.e., Mask R-CNN is 2.5 times faster than SegNet with an inference time of 0.66 s for a $256 \times 256$ pixel input image (Intel Core i7 with GeForce GTX 1080 Ti). The faster inference time observed for Mask R-CNN compared to SegNet is in line with other results from the literature [56].

### 7. Conclusions

We have presented a study on the robustness and generalization ability of popular network architectures for BFP extraction, i.e., SegNet as a representative for an encoder–decoder architecture and Mask R-CNN as a representative of an object-detector-based segmentation network. Therefore, our main focus was not on achieving peak performance compared to the state of the art but to evaluate strengths and weaknesses of the employed network architectures in different scenarios. The main findings of our study are as follows:

- Both network architectures can successfully benefit from additional training data and generalize well to the test data (RQ1).
- Both models generalize well on cities they did not see during training with only a marginal performance drop compared to previously seen cities (RQ2).
- While both models are well-suited for transfer learning settings (e.g., pretraining with additional external datasets), the effect of transfer learning seems to highly depend on the dataset characteristics (e.g., comparable resolution) and the network architecture (RQ3). There is no general trend across datasets and network models evaluated.
- Generalization ability across cities from different continents is high for both network models (RQ4). Large heterogeneity in the training set has shown to be beneficial in this scenario.
- The influence of different building structures (RQ5) and building densities (RQ6) in the training set has a strong effect on overall performance. A potential bias to certain building structures in the training data must not be underestimated. To achieve robust results, both network models need sufficient training examples with heterogeneous settlement structures.
- SegNet is more promising than Mask R-CNN, especially when training data is sparse. In the presence of large-scale training data (as, e.g., in the evaluation of RQ1), SegNet slightly outperforms Mask R-CNN. Nevertheless, it must be noted that SegNet provides pure segmentation, while Mask R-CNN additionally provides instance-level information.

With this study and our findings, we want to support other developers and researchers in creating powerful and, in particular, robust BFP extraction methods. We see several promising directions for extending our study in the future: (i) adding additional datasets for extended transfer learning experiments, (ii) evaluating additional network models to achieve a broader systematic comparison of approaches, (iii) integrating additional data modalities, such as infrared and depth information, to evaluate their beneficial effect, and (iv) adding data from additional continents to extend the investigation of generalization ability to the entire world.

**Author Contributions:** D.K., M.D. and M.Z. raised the funding for the research. All authors contributed to the conception and design of the study. Preparation of the material and data collection were carried out by M.S. The analysis was conducted by M.S. and E.S. Supervision was provided by M.Z. throughout all stages of the experiments. Additional analyses on the sensitivity and robustness of the approach as well as on runtime were performed by M.D., E.S. and M.Z. The first draft of the manuscript was written by M.S., D.K., M.D. and M.Z. E.S., M.D., D.K. and M.Z. contributed to the final draft. All authors have read and agreed to the published version of the manuscript.

**Funding:** This study was supported by the Austrian Research Promotion Agency (FFG), Project INFRABASE no. 865973. Special thanks go to GeoVille Information Systems and Data Processing GmbH for providing and preparing the GDBX dataset.

**Data Availability Statement:** The two benchmark datsets (INRIA and ISPRS) are publicly available, see Section 4. The GDBX data originates from company Maxar (formerly DigitalGlobe) and is commercially licensed and thus not publicly available.

**Conflicts of Interest:** All authors report no conflict of interest.

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
