# Peer review of "On the Robustness and Generalization Ability of Building Footprint Extraction on the Example of SegNet and Mask R-CNN"

_remotesensing, doi:10.3390/rs15082135_

Round 1
Reviewer 1 Report
Pixel-wise segmentation of buildings is still one of the most interesting topics in remote sensing. The submitted work investigates two popular deep learning based architectures (SegNet and Mask R-CNN) for the extraction of building footprints and evaluates their robustness in different conditions. Six main questions have been asked and then tried to be addressed in this research. By this, the authors tried to provide the readers with a fair and practical comparison of two different types of network architectures: encoder-decoder (SegNet as the representative) and object detection (R-CNN as the representative) following different experimental scenarios. Generally speaking, the topic, and let’s say, the concern of the submitted work is very interesting, to-the-point, and practical, as it provides valuable information to potential users of such networks for the same or any similar applications. On the other hand, the innovation level is not high (as it just combines well-known models and tools), but still acceptable. Finally, I suggest accepting the manuscript after addressing the following Major comment as well as some more minor suggestions.
Major:
- The experimental results and the comparisons are fairly complete and comprehensive. But I still suggest addressing the complexity of each model and reporting/comparing the computational time and other related parameters, preferably in a separated sub-section. That may enrich the article and increase its usability.
Minor:
- From the authors list it is not clear who the corresponding author is. Please add an asterisk to the name of the corresponding author in the list.
- Line 125, a comma is missing after ‘more recently’
- In Figure 2, the legend is missing for the blue layers in the Decoder part.
- In Figure 5, it looks better if the numbers 1 to 5 are overlaid on the images.
- In Table 4, I would suggest bolding the highest values for each case.
- In the caption of Table 4, some information is missing. For example, what do ‘Pr’ and ‘Re’ stand for? This should be stated clearly not only within the text but also in the caption. Also, in the paragraph above the table, it reads ‘Quantitative and qualitative results are summarized in Table 4. But isn’t it limited only to the quantitative results?
- The language is fine, but there are still some minor issues such as typos here and there that should be corrected.
Author Response
We thank the reviewer for the constructive feedback, which we believe helped us to improve the paper. Please find our responses to the individual points below in green.
Open Review
(x) I would not like to sign my review report
( ) I would like to sign my review report
English language and style
( ) English very difficult to understand/incomprehensible
( ) Extensive editing of English language and style required
( ) Moderate English changes required
(x) English language and style are fine/minor spell check required
( ) I don't feel qualified to judge about the English language and style
Concerning English language, we double checked the entire manuscript and corrected numerous spelling mistakes and typos.
Comments and Suggestions for Authors
Pixel-wise segmentation of buildings is still one of the most interesting topics in remote sensing. The submitted work investigates two popular deep learning based architectures (SegNet and Mask R-CNN) for the extraction of building footprints and evaluates their robustness in different conditions. Six main questions have been asked and then tried to be addressed in this research. By this, the authors tried to provide the readers with a fair and practical comparison of two different types of network architectures: encoder-decoder (SegNet as the representative) and object detection (R-CNN as the representative) following different experimental scenarios. Generally speaking, the topic, and let’s say, the concern of the submitted work is very interesting, to-the-point, and practical, as it provides valuable information to potential users of such networks for the same or any similar applications. On the other hand, the innovation level is not high (as it just combines well-known models and tools), but still acceptable. Finally, I suggest accepting the manuscript after addressing the following Major comment as well as some more minor suggestions.
We thank the reviewer for acknowledging our work
Major:
- The experimental results and the comparisons are fairly complete and comprehensive. But I still suggest addressing the complexity of each model and reporting/comparing the computational time and other related parameters, preferably in a separated sub-section. That may enrich the article and increase its usability.
We agree with the reviewer that complexity and computation time are important aspects for practical application. Especially, information on computation time was missing in the initial manuscript. We have added a section on computation time (training time and inference time) of both networks in Section 6.7 to provide a more complete picture.
Minor:
- From the authors list it is not clear who the corresponding author is. Please add an asterisk to the name of the corresponding author in the list.
We have added an asterisk for the corresponding author.
- Line 125, a comma is missing after ‘more recently’
Done.
- In Figure 2, the legend is missing for the blue layers in the Decoder part.
We have updated the legend and improved the quality of the figure.
- In Figure 5, it looks better if the numbers 1 to 5 are overlaid on the images.
We have assigned characters to the individual sub-images (a-e)
- In Table 4, I would suggest bolding the highest values for each case.
We agree that this makes the table better readable. It further better highlights the weakness of Mask R-CNN in terms of precision (which mainly explains its lower performance). We have added a respective remark in the caption of the table.
- In the caption of Table 4, some information is missing. For example, what do ‘Pr’ and ‘Re’ stand for? This should be stated clearly not only within the text but also in the caption. Also, in the paragraph above the table, it reads ‘Quantitative and qualitative results are summarized in Table 4. But isn’t it limited only to the quantitative results?
We have added explanations for the abbreviations in the caption and changed the text in the paragraph above. The word ‘Qualitative’ should refer to Figure 8, which we have made clearer now.
- The language is fine, but there are still some minor issues such as typos here and there that should be corrected.
We proof-read the entire paper and have corrected numerous typos and spelling mistakes.
Reviewer 2 Report
The motivation and significance of this article are not outstanding, and many research conclusions, especially RQ1, RQ2, and RQ3, etc., have been the general consensus in the field of remote sensing semantic segmentation research for several years ago.
Please clarify the contributions to this field.
The author mentioned that “we evaluate the robustness and generalizability of state-of-the-art methods as well as their transfer learning capabilities.”, while the experiments are conducted on the SegNet and Mask-RCNN both proposed in 2017. Their performance is much lower than many advanced networks such as the popular U-Net and YOLO.
Generally, a performance improvement of 1% on the evaluation metric is hard to gain. The precision preserved for the results which evaluated the performance of different methods in tables from 4 to 10 is too low. It can not compare the performance difference for related methods with a less than 1% accuracy gap.
The Mask-RCNN provides predicted masks and instance information simultaneously. It is recommended to provide the detection boxes for results extracted by Mask-RCNN for visualization. The results of Figures 10 and 11 contain very large buildings and fragmented buildings, which are not like those extracted by the instance segmentation model.
Author Response
We thank the reviewer for the constructive feedback, which we believe helped us to improve the paper. Please find our responses to the individual points below in green.
Open Review
( ) I would not like to sign my review report
(x) I would like to sign my review report
English language and style
( ) English very difficult to understand/incomprehensible
( ) Extensive editing of English language and style required
( ) Moderate English changes required
( ) English language and style are fine/minor spell check required
(x) I don't feel qualified to judge about the English language and style
Comments and Suggestions for Authors
The motivation and significance of this article are not outstanding, and many research conclusions, especially RQ1, RQ2, and RQ3, etc., have been the general consensus in the field of remote sensing semantic segmentation research for several years ago.
Please clarify the contributions to this field.
We agree that the aspects addressed by RQ1-3 have been investigated in the literature previously. When taking a closer look at the literature, there are, however, gaps that our study aims to close, see below. Additionally, we investigate more specific questions, i.e., RQ4-6, which have to our knowledge not been investigated in the literature. Ultimately, we could not find a detailed evaluation of the popular Mask R-CNN architecture for building footprint extraction, which makes our work complementary to existing works. Especially, with regard to RQ1-3 we want to clarify the particular contributions to the field in the following.
We agree with the reviewer that there exist publications in the remote sensing domain which investigate the impact of training set size (RQ1). Most of these works evaluate this aspect for rather traditional machine learning techniques (e.g. SVM, Random Forests, etc.) that utilize handcrafted features [1,2]. In [3] training set size is also investigated for two encoder-decoder models (SegNet and U-Net), showing that increasing training data facilitates performance. This work is, however, focusing on land cover segmentation and not in particular on building footprints. To the best of our knowledge, we are the first to directly compare an encoder-decoder based model (SegNet) with an object detection-based one (Mask R-CNN) with respect to training set size. Our results show that the impact of the training set size for these two popular models is consistent with the findings from previous investigations, which was so far not clear especially not for Mask R-CNN applied to building footprint segmentation.
We further agree with the reviewer that it is common to evaluate building footprint segmentation models on satellite images from unseen data/cities (RQ2) [4]. In the case of the public INRIA data set, for example, the complete test set is comprised of images from cities, which are not included in training and validation set. However, in our investigation for RQ2, we add a second test set, which like the training set consists of images from the same city but covering different locations. Comparing the metrics of this test set to the test set of completely unseen cities, we get direct insight to how attuned the network is to a distinct city structure. We can observe for example from Table 4 that the performance on unseen cities drops in most cases by a few percentage points. This is, however, different to the work in [5] where the authors report larger performance drops for unseen cities. Since domain adaption is often a challenge [5] and RQ2 has not been answered at least for Mask R-CNN, we included these experiments in the paper and we think that they represent useful and complementary information for practitioners and researchers in the field.
RQ3 focuses on transfer learning capabilities of the networks for building footprint extraction. Of course this has been investigated before but first on other models and second in a different way. In most publications that we could find on the topic, transfer learning refers to building upon models that were pretrained on either large image classification datasets like ImageNet [6-9] or segmentation datasets from other domains like COCO [10,11]. In our study, we investigate the effects of transfer learning from one satellite dataset to another one, i.e. a kind of intra-domain transfer learning. To the best of our knowledge this has not been investigated for the two network architectures before.
We have added a discussion to draw relations to existing literature for RQ1-3 in the paper.
[1] Boulila, Wadii. "A top-down approach for semantic segmentation of big remote sensing images." Earth Science Informatics 12 (2019): 295-306.
[2] Li, Manchun, et al. "A systematic comparison of different object-based classification techniques using high spatial resolution imagery in agricultural environments." International Journal of Applied Earth Observation and Geoinformation 49 (2016): 87-98.
[3] Ning, Huan, et al. "Choosing an appropriate training set size when using existing data to train neural networks for land cover segmentation." Annals of GIS 26.4 (2020): 329-342.
[4] Kaiser, Pascal, et al. "Learning aerial image segmentation from online maps." IEEE Transactions on Geoscience and Remote Sensing 55.11 (2017): 6054-6068.
[5] Wang, Rui, et al. "The poor generalization of deep convolutional networks to aerial imagery from new geographic locations: an empirical study with solar array detection." 2017 IEEE Applied Imagery Pattern Recognition Workshop (AIPR). IEEE, 2017.
[6] Borba, Philipe, et al. "Building Footprint Extraction Using Deep Learning Semantic Segmentation Techniques: Experiments and Results." 2021 IEEE International Geoscience and Remote Sensing Symposium IGARSS. IEEE, 2021.
[7] Safarov, Furkat, et al. "Improved Agricultural Field Segmentation in Satellite Imagery Using TL-ResUNet Architecture." Sensors 22.24 (2022): 9784.
[8] Alsabhan, Waleed, Turky Alotaiby, and Basil Dudin. "Detecting Buildings and Nonbuildings from Satellite Images Using U-Net." Computational Intelligence and Neuroscience 2022 (2022).
[9] Gao, Kyle, et al. "A region-based deep learning approach to instant segmentation of aerial orthoimagery for building rooftop detection, NRC Research Press"
[10] Luo, Liu, and Xiaoyi Guo. "Recognition and Extraction of Blue-roofed Houses in Remote Sensing Images based on Improved Mask-RCNN." International Core Journal of Engineering 8.2 (2022): 639-645.
[11] Chen, Shenglong, et al. "Large-Scale Building Footprint Extraction from Open-Sourced Satellite Imagery via Instance Segmentation Approach." IGARSS 2022-2022 IEEE International Geoscience and Remote Sensing Symposium. IEEE, 2022.
The author mentioned that “we evaluate the robustness and generalizability of state-of-the-art methods as well as their transfer learning capabilities.”, while the experiments are conducted on the SegNet and Mask-RCNN both proposed in 2017. Their performance is much lower than many advanced networks such as the popular U-Net and YOLO.
We have selected SegNet and Mask R-CNN for two reasons: First, because these networks represent popular backbones for many more advanced recent building footprint extraction approaches. For this reason, the original date of introduction of these methods was secondary to us. Second, because they represent two completely complementary approaches towards segmentation, i.e. SegNet is a one-stage segmentation net with an encoder-decoder architecture and Mask R-CNN is a two-stage segmentation network which first detects object bounding boxes and then segments them. We have improved the motivation and justification of method selection in the introduction.
We agree with the reviewer that U-Net (although it is even older, i.e., introduced 2015) would also be an interesting alternative to evaluate. Its benefit over SegNet is, however, not clearly proven in literature. While in some studies U-Net outperforms SegNet, in other it does not, e.g., in [12,14], where a performance difference of around 2% could be observed between SegNet and U-Net. Nevertheless, due to the similarity of SegNet and U-Net in architecture, we believe that many observations and conclusions from SegNet can be transferred to U-Net as well (e.g., the effect of additional training data, i.e., RQ1, which is supported by the study performed in [12]).
We further agree, that YOLO is a promising object detection architecture. We have investigated YOLO in depth in previous work, see e.g. [13]. We preferred Mask R-CNN over YOLO because (i) YOLO has – due to its pre-defined anchors – problems with the detection of very small objects, which is critical in the case of buildings and (ii) YOLO is – similarly to SegNet – a one-stage approach, which would limit our study to only one-stage approaches. Finally, YOLO itself does not provide a mechanism to obtain segmentation masks compared to Mask R-CNN, where boundary segmentation is directly integrated. Although segmentation extensions for YOLO exist, they are much less popular and frequently used than Mask R-CNN.
[12] Ning, H., Li, Z., Wang, C., & Yang, L. (2020). Choosing an appropriate training set size when using existing data to train neural networks for land cover segmentation. Annals of GIS, 26(4), 329-342.
[13] Kirchknopf, A., Slijepcevic, D., Wunderlich, I., Breiter, M., Traxler, J., & Zeppelzauer, M. (2022). Explaining YOLO: Leveraging Grad-CAM to Explain Object Detections. arXiv preprint arXiv:2211.12108.
[14] Khan, S. D., Alarabi, L., & Basalamah, S. (2023). An encoder–decoder deep learning framework for building footprints extraction from aerial imagery. Arabian Journal for Science and Engineering, 48(2), 1273-1284.
Generally, a performance improvement of 1% on the evaluation metric is hard to gain. The precision preserved for the results which evaluated the performance of different methods in tables from 4 to 10 is too low. It can not compare the performance difference for related methods with a less than 1% accuracy gap.
We agree with the reviewer that a certain level of precision is necessary to judge performance differences. We further agree that a performance difference with a less than 1% accuracy gap in our study is not indicative in the comparison of the two methods. We thus avoided to draw any conclusions from a performance difference lower that 1% and rather consider the methods to perform at the same performance level in such cases. In many cases in our experiments, however, we observe much stronger performance differences (>>1%). We believe that such results provide interesting insights on the performance of the approaches and represent a contribution to the community, especially because these two popular network architectures have so far not been compared directly in a head-to-head fashion.
Additionally, we want to point out that our study is deliberately designed to be open-ended and is primarily intended to identify trends, i.e., results were achieved without any specific tuning of one of the methods to provide an objective baseline. Our goal was to show how well the two methods perform on our dataset in different problem settings, without optimizing them further and thereby potentially over-fitting them to the data. Thereby, our main focus was not on achieving peak performance compared to the state-of-the-art but to evaluate strengths and weaknesses of the employed network architectures in different scenarios.
The problem of low precision is mainly present for Mask R-CNN, which can be seen in Table 4, leading to low DICE (i.e., F1) scores. This shows that Mask R-CNN cannot take benefit of additional training data in the same way as SegNet and further that the generalization to unseen cities is not as good as for SegNet. We could not find a direct comparison in the literature between SegNet and Mask R-CNN that highlights this performance difference and thus believe that these results are interesting for the community. We have extended our conclusions on these experiments in Section 6.2 and in Section 6.3 (where we also observe a strong performance difference between SegNet and Mask R-CNN).
The Mask-RCNN provides predicted masks and instance information simultaneously. It is recommended to provide the detection boxes for results extracted by Mask-RCNN for visualization. The results of Figures 10 and 11 contain very large buildings and fragmented buildings, which are not like those extracted by the instance segmentation model.
We agree with the reviewer, that such an analysis would be interesting to further analyse the performance of Mask R-CNN. For a direct comparison, however, we have primarily focused on the pixel segmentation outputs to enable a direct comparison between the methods. This is of course a certain limitation, which was necessary because both networks are very different in their architecture and in their segmentation capabilities (i.e., semantic segmentation in SegNet vs. instance segmentation in Mask R-CNN).
The results of Figures 10 and 11 show that Mask R-CNN (subfigures f-h) partly fails completely to segment large buildings. This is most probably a consequence of training Mask R-CNN on (a) rather small datasets (compared e.g., to experiments on RQ 1 and 2) and (b) data with a different building structure, i.e., small buildings, which leads to Mask R-CNN being biased to small bounding boxes that cannot capture larger buildings.
Reviewer 3 Report
The authors apply the known and widely applied methods. They try to answer six questions listed with doing experiements..
One question concerned is the rules of the categorization. How were they been defined, not clear? (They are Just before the line 284.)
The biggest lacking part of the paper is discussion. They should mention the previous works which they were focused on use of segnet and mask r-nn methods.
Secondly, the reason of selecting segnet and mask r-nn should be justified.
And last, the answers of the asked questions should be reviewed by the previous literature with a detailed analysis.
Author Response
We thank the reviewer for the constructive feedback, which we believe helped us to improve the paper. Please find our responses to the individual points below in green.
Open Review
(x) I would not like to sign my review report
( ) I would like to sign my review report
English language and style
( ) English very difficult to understand/incomprehensible
( ) Extensive editing of English language and style required
( ) Moderate English changes required
( ) English language and style are fine/minor spell check required
(x) I don't feel qualified to judge about the English language and style
Comments and Suggestions for Authors
The authors apply the known and widely applied methods. They try to answer six questions listed with doing experiements..
One question concerned is the rules of the categorization. How were they been defined, not clear? (They are Just before the line 284.)
These three subsets were automatically categorized using a simple heuristic building upon density estimation (number of buildings per area) and the building size (area covered by a building) derived from the ground truth masks. From all training patches we first compute the distribution of the number of building per patch. The heuristic is then applied on patch-level, i.e. each patch is categorized into one of the classes using the rules provided in the paper. To give an example: This means for example that patches where (i) the number of buildings is lower than the value corresponding to the 20th percentile of the distribution of building number across all patches and (ii) the area covered by buildings is higher than 15% of the total patch area are classified as sparse detached structure (SDS). We have added additional explanations and this example in the paper to make the rules behind this categorization more intuitive.
The biggest lacking part of the paper is discussion. They should mention the previous works which they were focused on use of segnet and mask r-nn methods.
We agree with the reviewer that this part was coming too short in the initial manuscript. We have added related work on SegNet and Mask R-CNN in Section 2 and we further made references and drew relations to results previously reported for SegNet and Mask R-CNN in the literature in Section 6 where we discuss the results for the individual research questions.
Secondly, the reason of selecting segnet and mask r-nn should be justified.
We have selected SegNet and Mask R-CNN for two reasons: First, because these networks represent popular backbones for many more advanced recent building footprint extraction approaches. Second, because they represent two completely complementary approaches towards segmentation, i.e. SegNet is a one-stage segmentation net with an encoder-decoder architecture and Mask R-CNN is a two-stage segmentation network which first detects object bounding boxes and then segments them. We have improved the motivation and justification for these methods in the introduction.
And last, the answers of the asked questions should be reviewed by the previous literature with a detailed analysis.
We agree that in particular the aspects addressed by RQ1-3 have been investigated in the literature previously. A closer look at the literature shows that there are gaps that our study aims to close. We have added a discussion including previous literature for RQ1-3 in the respective sections. RQ4-6 have not been studied in the literature to our knowledge. Furthermore, we could not find a detailed evaluation of the popular Mask R-CNN architecture for building footprint extraction in the context of RQ1-6, making our work complementary to existing work.
Round 2
Reviewer 2 Report
The title of the article should be revised to more accurately reflect the scope of the study, which primarily focuses on comparing the performance of SegNet and Mask-RCNN in building footprint extraction.
The article's contribution, particularly with regard to RQ1, may be limited as it addresses a commonly accepted challenge in computer vision and remote sensing applications.
The visual output of the building footprint extraction performed by Mask-RCNN appears unusual as all the recognized results are connected.
Author Response
Open Review
The title of the article should be revised to more accurately reflect the scope of the study, which primarily focuses on comparing the performance of SegNet and Mask-RCNN in building footprint extraction.
We agree with the reviewer and have adapted the title accordingly.
The article's contribution, particularly with regard to RQ1, may be limited as it addresses a commonly accepted challenge in computer vision and remote sensing applications.
We agree that the question in general has been investigated before (although not for Mask R-CNN in this context). We considered removing RQ1 for this reason, however, finally decided not to do it for the following reasons: First, in our research design, RQ1 and RQ2 are investigated within the same set of experiments. Removing RQ1 would also remove valuable results necessary to assess RQ2. Second, we consider RQ1 also as a sanity check that shows that the networks’ training process proceeds as expected and that they learn accurately. Third, the experiment shows an interesting pattern: for SegNet increasing training set size primarily facilitates recall while precision even slightly drops. For Mask R-CNN increasing training set size primarily facilitates precision, while recall drops slightly. This shows that Mask R-CNN detects buildings well, even with little training data and mainly suffers from false positive detections. SegNet, on the contrary, seems to be more robust to false positive detections but misses many buildings (false negatives) when the training set is small. This is against our expectation that more training data should improve both recall and precision and shows that both networks have individual and complementary strengths and weaknesses which - in theory at least - would make the two approaches promising candidates for combination
We have added these observations in the paper to provide more insights from the results and we further rephrased the research question to make it more general and more appropriate to our analysis.
The visual output of the building footprint extraction performed by Mask-RCNN appears unusual as all the recognized results are connected.
Attached buildings in our data are also attached in the ground truth. This is the reason why the networks tend to segment them as connected regions. In addition, for Mask R-CNN attached regions are also generated when bounding boxes largely overlap as e.g. in the results shown in Figure 10 and 11. This does, however, not mean that connected buildings in the result output are detected as one building.